# Long-term Variations in Ozone Levels in the Troposphere and Lower Stratosphere over Beijing: Observations and Model Simulations

Yuli ZHANG[1], Mengchu TAO[1,2], Jinqiang ZHANG[1], Yi LIU[1], Hongbin CHEN[1], Zhaonan CAI[1], Paul Konopka[2]

[1]Key Laboratory of Middle Atmosphere and Global Environment Observation, Institute of Atmospheric Physics, Chinese Academy of Sciences, Beijing 100029, China
[2]IEK-7: Stratosphere, Forschungszentrum Jülich, 52425 Jülich, Germany

*Correspondence to*: Y. Liu (liuyi@mail.iap.ac.cn)

**Abstract.** Tropospheric ozone is both a major pollutant and a short-lived greenhouse gas and has therefore attracted much concern in recent years. The ozone profile in the troposphere and lower stratosphere over Beijing has been observed since 2002 by ozonesondes developed by the Institute of Atmospheric Physics. Increasing concentrations of tropospheric ozone from 2002 to 2010 measured by these balloon-based observations have been reported previously. As more observations are now available, we used these data to analyze the long-term variability of ozone over Beijing during the whole period from 2002 to 2018. The ozonesondes measured increasing concentrations of ozone from 2002 to 2012 in both the troposphere and lower stratosphere. There was a sudden decrease in observed ozone between 2011 and 2012. After this decrease, the increasing trend in ozone concentrations slowed down, especially in the mid-troposphere, where the positive trend became neutral. We used the Chemical Lagrangian Model of the Stratosphere (CLaMS) to determine the influence of the transport of ozone from the stratosphere to the troposphere on the observed ozone profiles. CLaMS showed a weak increase in the contribution of stratospheric ozone before the decrease in 2011–2012 and a much more pronounced decrease after this time. Because there is no tropospheric chemistry in CLaMS, the sudden decrease simulated by CLaMS indicates that a smaller downward transport of ozone from the stratosphere after 2012 may explain a significant part of the observed decrease in ozone in the mid-troposphere and lower stratosphere. However, the influence of stratospheric ozone in the lower troposphere is negligible in CLaMS and the hiatus in the positive trend after 2012 can be attributed to a reduction in ozone precursors as a result of stronger pollution control measures in Beijing.

## 1 Introduction

Tropospheric ozone is an important pollutant and is detrimental to both human health (WHO, 2006) and the productivity of vegetation (Ainsworth et al., 2012; Emberson et al., 2013; Feng et al., 2015). It is also an important greenhouse gas (IPCC, 2007) and influences radiative forcing (Wang et al., 1976; Lacis et al., 1990; Seinfeld et al., 2006). It is therefore crucial to understand and monitor the long-term variability in tropospheric ozone.

Tropospheric ozone mainly originates from photochemical reactions involving precursors such as nitrogen oxides ($NO_x$) and volatile organic compounds (VOCs) (Monks et al., 2009; Su et al., 2018; Tan et al., 2018). The exchange of ozone between the stratosphere and the troposphere is also important to bring ozone into troposphere (Dufour et al., 2010, 2015; Neu et al., 2014).

Human activities have significantly increased tropospheric ozone since the industrial revolution as a result of increased concentrations of ozone precursors (Hough and Derwent, 1990; Parrish et al., 2012). The reduction in surface UV radiation due to high aerosol concentrations has important impacts on the production of photochemical ozone (Deng et al., 2011). Studies have shown a dramatic positive trend in the concentration of tropospheric ozone in China since the 1990s due to rapid economic development and urbanization (Wang et al., 2012; Cooper et al., 2014; Chen et al., 2015; Verstraeten et al., 2015). Increasing ozone concentrations have been observed both at the surface (Cooper et al., 2014; Ma et al., 2016; Wang et al., 2009a) and in the lower troposphere (Ding et al., 2008; Shen et al., 2012; Sun et al., 2016; Wang et al., 2017).

Air quality controls have been implemented in China as a result of the recent increases in atmospheric pollutants, especially in the North China Plain, and the emissions of $SO_2$ and $NO_x$ have been successfully reduced in recent years (Ma et al., 2016; van der A et al., 2017; Li et al., 2017). Recent studies have shown levelling off/decrease in surface ozone levels in rural areas of eastern China and in outflow of eastern China air masses (Xu et al., 2020; Wang et al., 2019). Using the Infrared Atmospheric Sounding Interferometer (IASI) onboard the European Space Agency's (ESA) MetOp series of polar orbiting satellites, together with surface and ozonesonde measurements, Dufour et al. (2018) reported the trend in tropospheric ozone concentrations over the North China Plain for the time period 2008–2016. They found that there were two distinct periods: (1) 2008–2012 with no significant trend; and (2) 2013–2016 with a significant negative trend in lower tropospheric ozone.

Ozone sounding has been carried out over Beijing on a regular basis since 2002 and is the longest observation period of the ozone profile over the North China Plain (Zhang et al., 2014). As a unique long-term series of ground-based observations, this dataset is the best candidate with which to reveal the long-term variability in tropospheric and lower stratospheric ozone over the North China Plain and especially over Beijing. Wang et al. (2012) used this dataset to show a positive trend in tropospheric ozone during the time period 2002–2010. As an extension of the work by Wang et. al. (2012), we got more years of data since 2010. In this work, we used the whole time series (2002–2018) of ozonesonde observations to explore the variability in ozone concentrations over Beijing in the last two decades.

## 2 Data and model

### 2.1 Ozonesonde

Ozone concentrations from the Earth's surface up to ~30 km were measured over Beijing using an ozonesonde developed by the Key Laboratory of Middle Atmosphere and Global Environment Observation of the Institute of Atmospheric Physics (IAP) (Zhang et al., 2014). The IAP ozonesonde is based on an electrochemical method which is well documented in previous studies (Wang et al., 2003; Xuan et al., 2004; Zheng and Li, 2005). The ozonesonde has previously been compared with the widely used electrochemical concentration cell (ECC) developed by Komhyr (1969) and the Brewer spectrophotometer (Zhang et al., 2014) and was able to capture the ozone profile. The mean difference in the ozone partial pressure between the IAP and ECC ozonesondes was <0.5 mPa in the troposphere and <1 mPa in the lower stratosphere. The correlation coefficients for profiles by IAP ozonesondes and the ECC are greater than 0.99 (Xuan et al., 2004). The total ozone columns measured by the IAP ozonesonde and the Brewer spectrophotometer were in good agreement with a relative difference of 6%. For the total ozone column, the relative difference and correlation coefficient between IAP ozonesonde and Brewer

instrument were 6% and 0.94. The ozonesonde data has been used to validate satellite measurements (Bian et al.,
2007) and model products (Wang et al., 2012).

The ozone profiles have been observed about once a week since 2002 at 14:00 local time (06:00 UTC). In some intensive observation periods (e.g., 24 March to 10 April 2003), ozonesondes were launched every day. However, there was no observation (gaps in Figure 1) in two periods (July 2008 and January 2013). The ozonesondes were released from Beijing Observatory (39.81°N, 116.47°E; 31 m above sea level). The maximum altitude for the ozonesonde profile, which depends on the altitude at which the balloons burst, is between 25 and 35km.

**2.2 Chemical transport model**

The Chemical Lagrangian Model of the Stratosphere (CLaMS v1.0) chemical transport model was used to quantify the variation in tropospheric ozone caused by transport from the stratosphere. The CLaMS contains a comprehensive set of reactions of relevance to the stratosphere, including full chlorine and bromine chemistry, 36 chemical species, and 115 reactions (including 27 photolysis and 11 heterogeneous reactions) (McKenna et al., 2002). The model chemistry integrations are based on A Selfcontained Atmospheric Chemistry coDe (ASAD) Carver et al. (1997). The chemical reactions (gas phase and photolysis) are summarized by McKenna et al. (2012). Based on a Lagrangian representation, CLaMS is well suited to simulations of tracer transport (McKenna et al., 2002, Konopka et al., 2004, 2019).

We used the 40-year CLaMS transient simulation starting on 1 January 1979 and driven by horizontal winds and the diabatic heating rates (vertical velocities) derived from the ERA-Interim reanalysis (Dee et al., 2011). The configuration and model initialization followed the model setup described in Wang et al. (2012) and Pommrich et al. (2014) (100 km horizontal/400 m vertical resolution around 380 K). The first 10 years of the CLaMS transient simulation can be considered as the model spin-up time. To isolate and quantify the long-term trend caused by transport from the stratosphere, a CLaMS simulation without ozone chemistry in troposphere is considered. The ozone values in the lowest model layer were set to zero. CLaMS comprises three main modules: Lagrangian advection; mixing; and stratospheric chemistry. Because there is no tropospheric photochemistry in CLaMS, the tropospheric ozone simulated by CLaMS mainly descends from the stratosphere. The daily output of CLaMS was interpolated at the ozonesonde locations in Beijing for all observed profiles.

**2.3 Nitrogen Dioxide from OMI**

To discuss the long-term variation of tropospheric ozone precursor in Beijing, we use the version 3 of Aura Ozone Monitoring Instrument (OMI) Nitrogen Dioxide ($NO_2$) standard product (Krotkov et al., 2018). $NO_2$ is an important chemical species in troposphere where it is a precursor to ozone production. OMI is a contribution of the Netherlands's Agency for Aerospace Programs (NIVR) in collaboration with the Finnish Meteorological Institute (FMI) to the Aura mission. The Earth is viewed in 740 wavelength bands along the satellite track with a swath large enough to provide global coverage in 14 orbits (1 day). Due to its unprecedented spatial resolution and daily global coverage, OMI plays a unique role in measuring trace gases important for the ozone layer, air quality, and climate change (Levelt et al., 2018). It measures the total ozone and other atmospheric parameters related to ozone chemistry and climate such as $NO_2$, $SO_2$, and aerosols. In this study, we select OMI tropospheric columns of $NO_2$ one degree around Beijing.

## 3 Comparison between the ozonesonde data and CLaMS simulation

The concentration of tropospheric ozone has strong seasonal variations with a minimum in winter and a maximum in summer. To better estimate the ozone trend, we calculated the contribution of each month to the annual $O_3$ to remove the seasonality from the time series, leaving the deseasonalized ozone data. Figure 1 therefore shows the deseasonalized ozone mixing ratio measured by the IAP ozonesondes and simulated by CLaMS in the troposphere and lower stratosphere during the time period 2002–2018. The ozonesonde observations (Figure 1a) show that the concentrations of tropospheric ozone increased in the period 2002–2012. This positive trend is consistent with the study of Wang et al. (2012) in which the same ozonesonde data were used. They suggested, in agreement with other studies (Wang et al., 2006; Wang et al., 2009b; Chou et al., 2009), that photochemical ozone production is the primary reason for the increase in ozone concentrations in the troposphere. A sudden decrease in the ozone mixing ratio occurred in the upper troposphere and lower stratosphere (UTLS) from late 2011 to early 2012. The concentrations of tropospheric ozone have not increased since 2012. The tropospheric ozone mixing ratios have remained stable and are almost equal to the levels in 2005–2006.

The ozone mixing ratio simulated by CLaMS (Figure 1b) captures the main characteristics of the deseasonalized ozonesonde observations in the mid-troposphere (3–9 km) and the UTLS (9–15 km). In particular, the CLaMS ozone mixing ratio below 10 km in the period 2009–2012 was larger than the ratio before and after this period. However, the CLaMS simulations in the lower troposphere (0–3 km) were much smaller than the observations.

To quantify the differences between the ozonesonde measurements and the CLaMS simulations, Figure 2 shows the correlations between the respective partial columns (9–15, 3–9 and 0–3 km) in four seasons (winter: December–January–February; spring: March–April–May, summer: June–July–August and autumn: September–October–November). In the lower troposphere, the ozone columns simulated by CLaMS were much smaller than the those measured by the ozonesonde (Figure 2c). This is because there is no tropospheric ozone chemistry, the main source of ozone in the lower troposphere, in CLaMS (Monks et al., 2009). In the mid-troposphere, transport from stratosphere is the main source of CLaMS ozone because of the lack of tropospheric ozone chemistry in the model. The CLaMS simulations in the mid-troposphere are much closer to the ozonesonde measurements (Figure 2b). CLaMS overestimates the transport of ozone from the stratosphere to the troposphere, which is strongest in spring. This is because CLaMS has deficiencies in the representation of the effects of convective uplift and mixing due to weak vertical stability in the troposphere (Konopka et al., 2019). The destruction of ozone in the mid-troposphere is not completely included in the model as a result of the absence of tropospheric ozone chemistry. A larger production of ozone is expected as a result of reactions involving water vapor, hydrogen peroxy and hydroxyl radicals (Stevenson et al., 2006). The CLaMS simulations in the UTLS agree well with the ozonesonde observations and only slightly overestimate the observations in spring (Figure 2a).

## 4 Long-term variations in ozone concentrations

To quantify the long-term variations in ozone concentrations, the deseasonalized partial column of ozone observed by the ozonesonde and simulated by CLaMS are shown in Figures 3 and 4 for the lower troposphere, mid-troposphere and UTLS. Figure 6 explores the seasonal dependence of the respective partial columns and their trends. Based on the ozonesonde observations, two features determine the variations in ozone concentrations after 2012: (1) the sudden decrease in ozone from late 2011 to early 2012; and (2) the hiatus in the positive trend after

2012. The sudden decrease is defined as the period in which the most significant decrease in Gaussian-weighted deseasonalized ozone was observed (the period between the two blue dashed lines).

The variations in the precursors of tropospheric ozone have dominant roles in the long-term variability of tropospheric ozone. In recent years, the Chinese government has started to invest time and resources in controlling air pollution. A review of 20 years of air pollution control in Beijing (UN Environment, 2019) reported reductions in $NO_x$ during the period 2013–2017. A clear decreasing trend in $NO_x$ emissions has been observed since 2012 (van der A et al., 2017). Zheng et al. (2018) also reported that emissions of $NO_x$ in China decreased by 21% during the time period 2013–2017. Wang et al. (2019) reported that $NO_x$ emissions in eastern China decreased by ∼25% from 2012 to 2016. Tropospheric $NO_2$, one of the precursors of tropospheric ozone, has gradually decreased over Beijing in recent years (Vu et al. 2019).

We use the tropospheric column of $NO_2$ from OMI to discuss the influence of precursors on the long-term variation of tropospheric ozone in Beijing. The deseasonalized tropospheric columns of $NO_2$ measured by OMI from 2004 to 2018 are shown in Figure 5. Tropospheric $NO_2$ was increasing from 2004 to 2010, especially in 2009, leading to the increase of ozone in lower and upper troposphere. As Chinese government start to control air pollutions, tropospheric columns of $NO_2$ were in a condition of relatively large fluctuation in the period of 2010-2013. Tropospheric $NO_2$ over Beijing experienced two major fluctuations in this period, as shown by Gaussian-weighted means. Then tropospheric $NO_2$ was gradually decrease since 2013, result in the hiatus of ozone increase in lower and upper troposphere.

Even if a massive reduction in the precursors of tropospheric ozone results in a local decrease in ozone in the lower troposphere, the transport of ozone from the stratosphere to the troposphere is widely considered to be an important source of tropospheric ozone. We used the CLaMS simulations to show the role of the transport of ozone from the stratosphere in modulating the concentration of tropospheric ozone (Figure 4). Because there is no photochemical reaction in the troposphere in CLaMS, the simulated variations in tropospheric ozone can only be transported from the stratosphere. As a result, the ozone columns in the lower troposphere simulated by CLaMS (Figure 4c) are much smaller than those measured by the ozonesonde (Figure 3c). The observed sudden decrease in ozone in the mid-troposphere and UTLS is also clear in the CLaMS simulations, which means that this decrease originates in the stratosphere. Because the decrease in ozone in the stratosphere and the start of pollution control measures in Beijing occurred at roughly the same time, we have to separate the two variations to understand the trends in ozone concentrations. An exploration of the seasonal dependence of the respective partial columns (Figure 6) will help to narrow down this problem.

Tropospheric ozone chemistry dominates the trends in the lower troposphere in summer and autumn. The contribution in CLaMS is so small here that any stratospheric influence can be neglected. We call this range the "troposphere-dominated range". By contrast, the stratospheric influence is dominant in the UTLS in winter and spring and the tropospheric contribution can be ignored. We call this range the "stratosphere-dominated range". All the other combinations of seasons and altitudes are a superposition of the troposphere- and stratosphere-dominated ranges and we call such combinations the "superposition range".

The sudden decrease in ozone concentrations in the troposphere-dominated range in 2011–2012 ended the positive ozone trend that had been observed since 2002 and was most prominent in lower troposphere during summer and autumn (Figure 6g and 6j). The trend after the decrease was almost neutral, indicating that air pollution control measures effectively reduced the concentration of ozone in the lower troposphere. In the stratosphere-dominated

range, both the ozone trends observed by the ozonesonde and those simulated by CLaMS became negative after 2013, most clearly during winter and spring in UTLS (Figure 6c and 6f). In the superposition range, most of the observed increasing trends weakened rapidly after 2012 (all other panels in Figure 6). Almost all the trends simulated by CLaMS became negative. The increasing trend simulated in the mid-troposphere by CLaMS in spring was almost parallel to the trend observed by the ozonesonde before the decrease (Figure 6e). However, the trend simulated by CLaMS became negative after the decrease when the trend observed by the ozonesonde was still slightly positive due to the upward influence of the positive trend in the lower troposphere. Thus changes in the trends in the superposition range can only be understood as an interaction between the impact of air pollution control and the changing influence of the stratosphere.

Table 1 summarizes the (linear) ozone trends before and after 2012 in ozone concentrations calculated for all four seasons. The first column gives the mean values of ozone ($M_{O3}$, units: DU) observed by the ozonesonde and simulated by CLaMS. All the numbers are broken down to the altitude range and season considered. The ozone trends ($T_{O3}$) are calculated in DU/year and the relative ozone trends ($T_{rel}$) are defined as the percentage of $T_{O3}$ in $M_{O3}$, i.e., $T_{rel} = T_{O3}/M_{O3}$ (%).

The change in the ozone trend ($\Delta_{rel}$) was calculated as the difference between the relative trends after and before the decrease, i.e., $\Delta_{rel}$ (%) = $T_{rel}$ (after the decrease) − $T_{rel}$ (before the decrease). For example, in the lower troposphere, where the most significant change in ozone occurred in the autumn, the mean concentration of ozone was 13.18 DU. The trend was 3.34 DU/year (25.3% relative to 13.18 DU) before the decrease and −0.33 DU/year (−2.5% relative to 13.18 DU) after the decrease. The reversal of the relative trend can therefore be quantified as the absolute value of $\Delta_{rel}$, i.e., $|\Delta_{rel}| = |-2.5\% - 25.3\%| = 27.9\%$.

In this paper, we only discuss time evolution of the ozone columns with absolute values of $\Delta_{rel}$ >20%, which typically describe the reversal in trend from positive before 2012 to neutral or negative after 2012. Thus, the most apparent change in the lower troposphere occurred in autumn (27.9%). In the mid-troposphere, the largest changes in trend were observed in spring (19.6% by ozonesonde and 47.7% by CLaMS) and autumn (27.4% by ozonesonde and 36.2% by CLaMS). In the UTLS, the highest values were observed in spring (23.0 and 34.2% by ozonesondes and CLaMS, respectively).

**5 Quantified ozone trends**

The quantified trends show us how significant the variations in ozone were over Beijing during the time period 2002–2018. The trend after 2012, the period and the magnitudes of the decrease varied with the atmospheric layers (Figure 3). The ozone columns measured by ozonesonde in the lower troposphere showed an abrupt decrease of 5 DU from late 2011 to mid-2012, which is estimated by the Gaussian-weighted means of the deseasonalized ozone concentrations (red curves). The increasing trends, estimated as 0.06 DU/month before the decrease, slowed to 0.02 DU/month (Figure 3c). A similar magnitude of decrease occurred in the mid-troposphere from mid-2011 to 2012 (Figure 3b). The hiatus in the increase of ozone was more pronounced at this level. The ozone column in this level showed a rate of increase of 0.08 DU/month before the decrease and this positive trend became neutral (0 DU/month) after the decrease. The mean ozone columns in both the lower and mid-troposphere after 2012 returned to levels almost equal to the columns in 2005–2006. The decrease in the ozone column in the UTLS (Figure 3a) occurred in a period from mid-2011 to the end of 2011 and the ozone levels recovered to those

observed before the decrease in early 2012. The increasing trend of 0.05 DU/month became neutral (0 DU/month) after the decrease.

The ozone trend simulated by CLaMS was almost zero (-0.01 DU/month) in the UTLS until a sudden decrease in 2011. The concentration of ozone in the UTLS then decreased at a rate of −0.11 DU/month. In the mid-troposphere, where ozone is a result of downward transport, the increasing trend was 0.05 DU/month before the decrease (Figure 4b). This increasing trend was slower than its counterpart measured by the ozonesonde (0.08 DU/month) as a result of the absence of tropospheric ozone photochemistry in CLaMS. For the same reason, instead of remaining neutral as in the ozonesonde measurements (Figure 3b), the CLaMS ozone in the mid-troposphere decreased at a faster rate of −0.07 DU/month after the decrease in 2011 (Figure 4b). The ozone columns simulated by CLaMS in the lower troposphere (Figure 4c) were much smaller than those measured by the ozonesonde (Figure 3c) as a result of the absence of tropospheric photochemistry. This comparison indicates that the decrease in ozone and the hiatus in the increase of ozone in this troposphere-dominated range is the result of the air pollution control measures started by the Chinese government.

The increasing trends in the troposphere-dominated range became slower after the decrease in 2011–2012 ($\Delta_{rel}$ = −17.3% in summer and −27.9% in autumn). The trends changed from 3.91 to 0.39 DU/year in summer and from 3.34 to −0.33 DU/year in autumn, proving the influence of air pollution control measures on the hiatus in the increase in ozone. The positive ozone trends observed by ozonesondes in the stratosphere-dominated range became negative, with $\Delta_{rel}$ = −9.6% in winter and −23.0% in spring. The changes in the trends simulated by CLaMS were more dramatic ($\Delta_{rel}$ = −23.0% in winter and −34.2% in spring).

Most of increasing trends weakened rapidly (negative values of $\Delta_{rel}$) in the interaction range or even began to decrease after 2012, except for the trend in the lower troposphere in winter, which increased by 9.4%. Of all the interaction ranges, the most significant change in ozone concentrations occurred in the mid-troposphere in spring and autumn. These trends observed by ozonesondes changed by about 5 DU/year with $\Delta_{rel}$ = −19.6% in spring and −27.4% in autumn. Their counterparts simulated by CLaMS changed even more dramatically, with $\Delta_{rel}$ = −47.7% in spring and −36.2% in autumn. All the trends simulated by CLaMS became negative, except in winter in the mid-troposphere when the trend was 0.67 DU/year. However, although the trends decreased a lot after the sudden decrease in ozone concentrations, most of the actual trends simulated by ozonesonde were still positive. These comparisons between the ozonesonde and CLaMS results shows that the concentrations of ozone should have decreased since 2012 if we only consider the influence of transport from the stratosphere. However, the actual trends after 2012 in the mid-troposphere did not decrease as much as in CLaMS because they were affected by the much slower, but still increasing or almost neutral, trends in the lower troposphere.

In general, the quantified ozone trends revealed the hiatus in the increase in ozone over Beijing. Although influenced by chemistry and stratospheric transport, the changes in the ozone trend varied with both altitude and season. In most situations, the increase in ozone has been moderated since 2012.

**6 Discussion and conclusions**

We observed tropospheric and lower stratospheric ozone columns in Beijing once a week from 2012 using ozonesondes developed by the IAP. Using these data, Wang et al. (2012) found a positive ozone trend during the time period 2002–2010. We extended these data to 2018 and found that the evolution of this trend after 2010 was

strongly determined by two factors: (1) a sudden decrease in mainly stratospheric ozone from late 2011 to early 2012; and (2) a decrease in mainly tropospheric ozone caused by reduction in air pollution in the Beijing region. The Chinese government has taken action to reduce air pollution since 2012 and the precursors of ozone have decreased gradually in recent years (Vu et al., 2019; Zheng et al., 2018). We show the reduction in tropospheric $NO_2$ by using OMI measurements. Other studies have also shown that the other $O_3$ precursors have decreased in

recent years in China, including not only NOx but also SO2 and VOCs (Ma et al., 2016; van der A et al., 2017; Li et al., 2017; UN Environment, 2019; Wang et al., 2019). These reduction in ozone precursors are considered to be the main reason for the hiatus in the increase in ozone in the troposphere, especially in the lower troposphere. The decrease in stratospheric ozone has a more global origin. Chen et al. (2019) investigated the long-term variation (1979-2016) of tropopause in China by using the newly released quality-controlled radiosonde data from

China Meteorological Administration. The result shows an upwards trend of tropopause in most part of China including the North China Plain. The uplifted tropopause may result in the reduction of ozone in UTLS. Diallo et al. (2019) reported that a strong La Niña phase of the El Niño–Southern Oscillation around 2011 caused an anomalous increase in ozone in the lower stratosphere of the tropics. The related weaker upwelling in the tropics coincided with weaker downwelling in extra-tropical regions and, as a consequence, less transport of ozone from

the stratosphere to the troposphere, especially at the latitude of Beijing. Because there is no tropospheric ozone chemistry in CLaMS, this is clearly manifested as a decrease in ozone in the CLaMS simulation interpolated along the ozonesonde profiles (Figure 4a and 4b). The observed sudden decrease of ozone in the mid-troposphere and UTLS in the period of 2011-2012 is also clear in the CLaMS simulations, which means that this sudden decrease originates in the stratosphere. The CLaMS simulations were much closer to the ozonesonde measurements in the

mid-troposphere, but CLaMS seemed to overestimate the transport of ozone from the stratosphere to the troposphere in the UTLS.

    Because pollution control measures in Beijing began around the same time as the decrease in stratospheric ozone, it is difficult to separate quantitatively their contribution to the observed reversal in trend. However, it is possible, at least qualitatively, to separate two ranges using CLaMS: (1) a tropospheric range in which the influence of

stratospheric ozone is negligible; and (2) a stratospheric range where this influence is substantial. Thus, the control of air pollution in the troposphere-dominated range effectively reduced the concentration of ozone after 2012. By contrast, the ozone trends in the stratosphere-dominated range observed by the ozonesondes or simulated by CLaMS became zero or negative after 2012. In the superposition range, most of the observed increasing trends weakened rapidly after 2012. Almost all the trends simulated by CLaMS became negative. The changes in the

trends in the superposition range can be understood as an interaction between the impact of air pollution control and the changing influence of the stratosphere.

    We conclude that the abrupt decrease and the deceleration of the increase in ozone in the troposphere and lower stratosphere is mainly the result of two overlapping effects: (1) the environmental protection measures implemented in recent years; and (2) variations in the transport of ozone from the stratosphere. Although the

reduction in tropospheric ozone precursors played a dominant part, the effect of the transport of ozone from the stratosphere to the troposphere should not be ignored. Recently, there are studies indicate the surface ozone trends are affected by meteorological variation. Liu et al. (2020) recently found that higher temperatures after 2013 led to an increase in O3 concentrations in northern China via an increase in biogenic emissions. They assessed the effects of changes in meteorology (temperature, specific humidity, wind field, planetary boundary layer height,

clouds, and precipitation) on ozone levels. Li et al. (2019) indicate that an important factor for ozone trends in the North China Plain is the ~40% decrease of fine particulate matter (PM2.5) over the 2013–2017 period. More observations are needed to investigate the variations in tropospheric ozone precursors and in related meteorology to fully understand the long-term variations in tropospheric and lower stratospheric ozone. More details of the transport of ozone from the stratosphere are expected to be revealed by comparing the CLaMS simulations of

ozone with other observations. The observations of ozone over Beijing by IAP ozonesondes will be continued and we expect more improvements in reducing tropospheric ozone pollution.

**Acknowledgments**

This work was supported by the Strategic Priority Research Program of Chinese Academy of Sciences (Grant No. XDA17010105). This research was also supported by the National Key R&D Program of China

(2017YFC0212800).

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

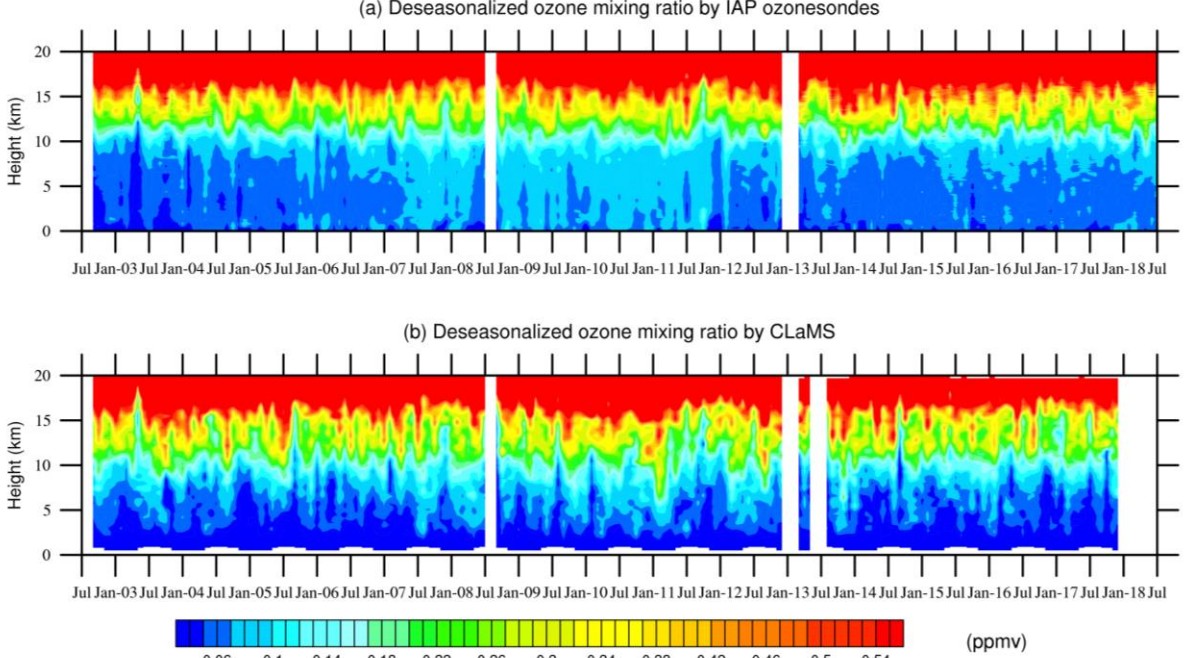

**Figure 1. Deseasonalized monthly mean ozone mixing ratio (units: ppmv) over Beijing (a) measured by the IAP ozonesonde and (b) simulated by CLaMS. There was no ozonesonde observation in July 2008 and January 2013. CLaMS is unable to calculate ozone mixing ratio in July 2013, because the information of balloon locations were lost during this period.**

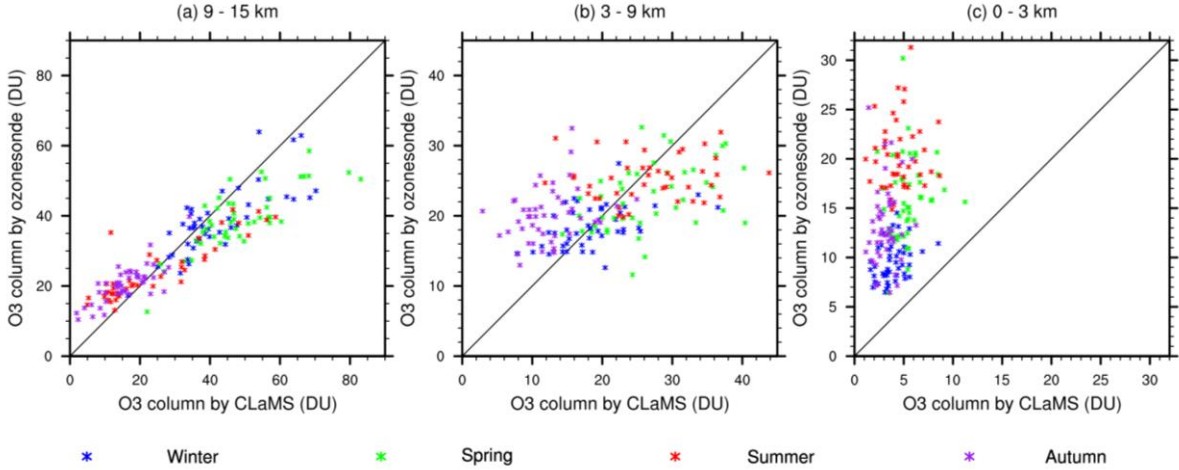

**Figure 2. Validation of ozone columns simulated by CLaMS (in Dobson units, DU) by comparison with the IAP ozonesonde in (a) the UTLS, (b) the mid-troposphere and (c) the lower troposphere during 2012–2018. Each point represents the average of one month of measurements. Measurements in different seasons (winter: December– January–February; spring: March–April–May, summer: June–July–August and autumn: September–October– November) are shown as different colors.**

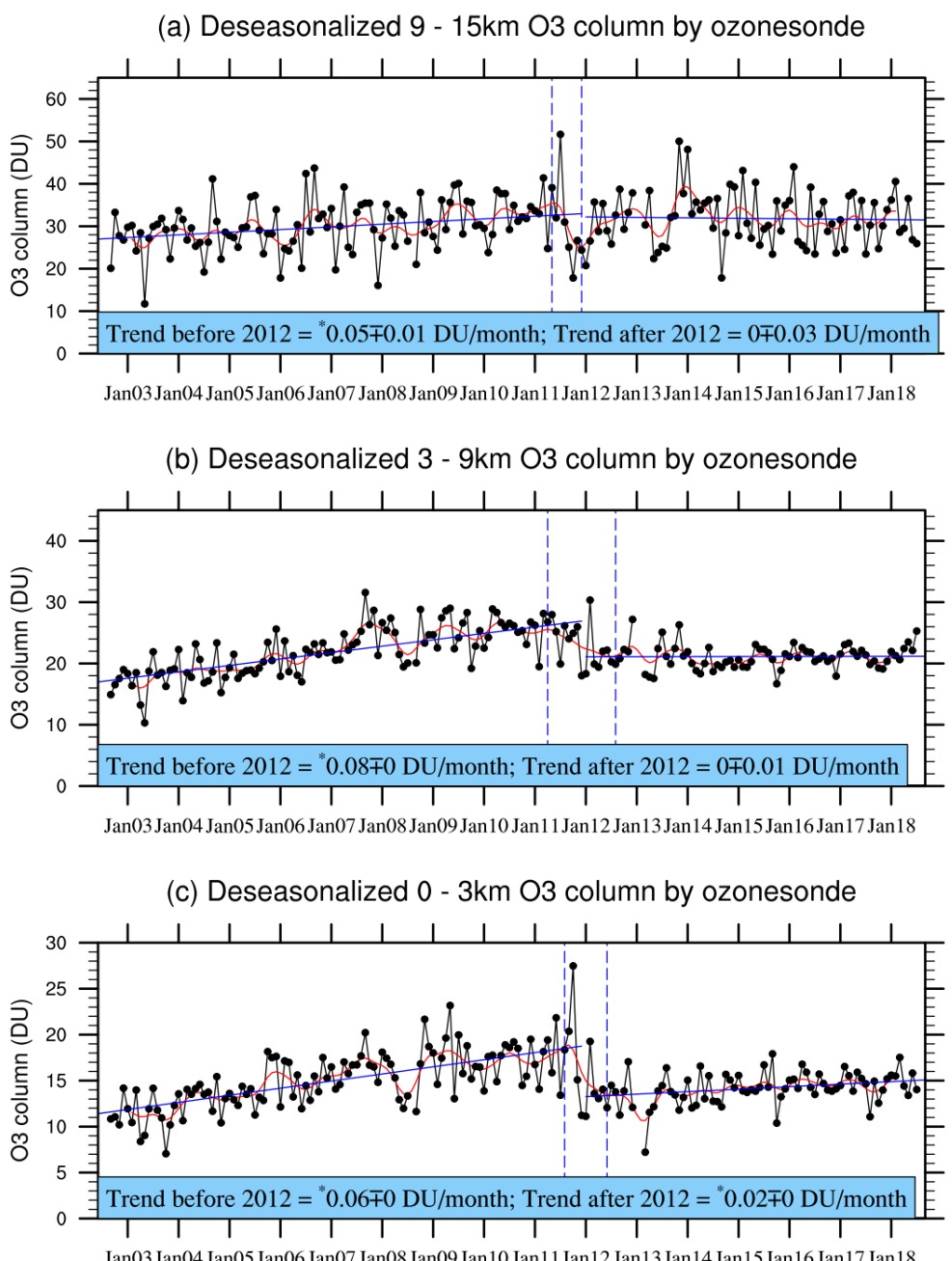

**Figure 3. Deseasonalized monthly mean partial columns of ozone over Beijing (black solid lines and dots) measured by the IAP ozonesonde and the corresponding Gaussian-weighted means using a half-width of 12 months (red curves). The blue solid lines estimate the linear trends (slope ∓ standard error) before and after the decrease in ozone during late 2011 and early 2012. The periods of decrease are represented by the time between the two blue dashed lines. The trends with "*" passed the 95% significance criterion.**

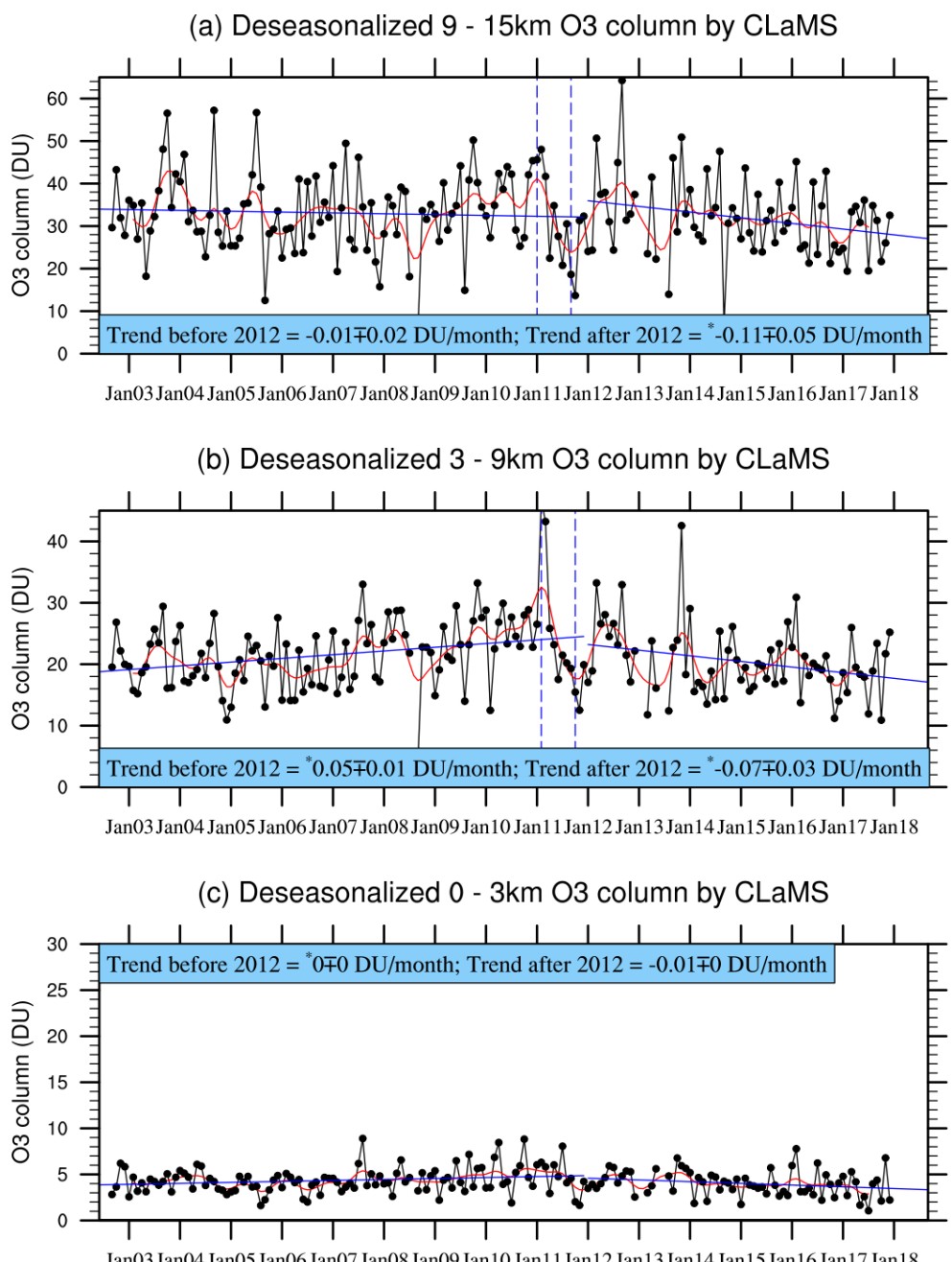


**Figure 4. Deseasonalized monthly mean partial columns of ozone over Beijing (black solid lines and dots) simulated by CLaMS and the corresponding Gaussian-weighted means using a half-width of 12 months (red curves). The blue solid lines estimate the linear trends (slope ∓ standard error) before and after the decrease in ozone during late 2011 and early 2012. The periods of decrease are represented by the time between the two blue dashed lines. The trends with "*"**

**passed the 95% significance criterion.**

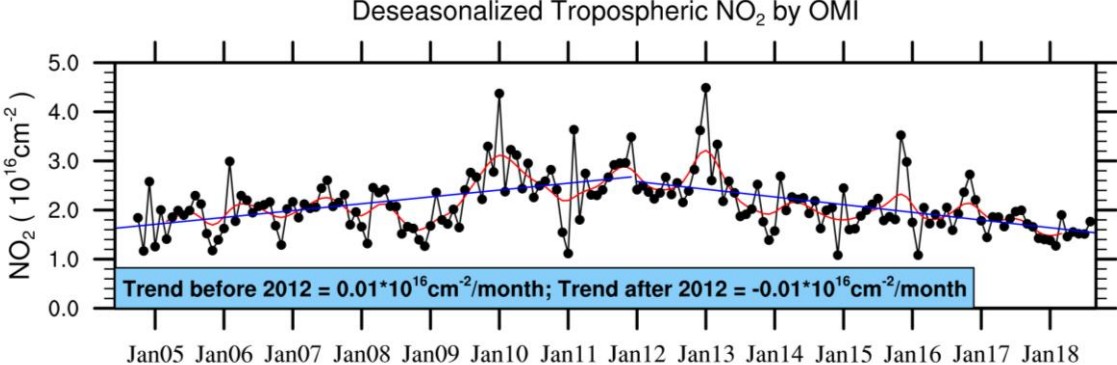

**Figure 5. Deseasonalized monthly mean columns of tropospheric NO₂ (black solid lines and dots) and the corresponding Gaussian-weighted means using a half width of 12 months (red curves). The blue lines estimate the linear trend before and after the sudden decrease of ozone during late 2011 and early 2012.**


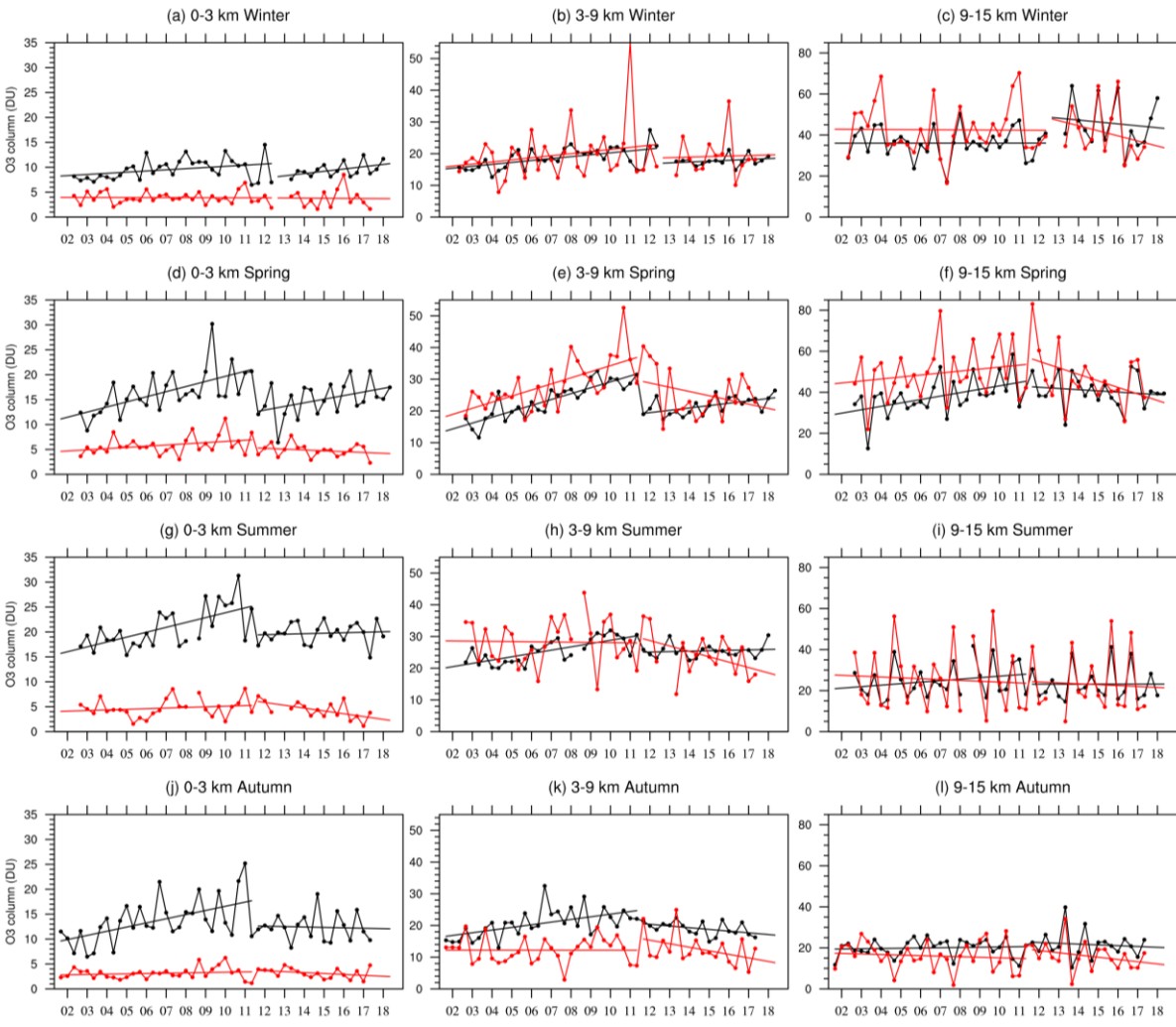

**Figure 6. Monthly mean column ozone (DU) from the ozonesonde observations (black) and CLaMS simulations (red) in four seasons. Trends of column are calculated before and after the sudden decrease of ozone in 2011-2012. There are 3 monthly values in each year for each season.**

**Table 1. Trends in the monthly mean partial column ozone at three altitudes in four seasons. The trends with "*" passed the 95% significance criterion. The trends with "※" passed the 90% significance criterion.**

| | | Ozonesonde | | | | | | CLaMS | | | | |
|---|---|---|---|---|---|---|---|---|---|---|---|---|
| | | $M_{O3}$ (DU) | Before 2012 | | After 2012 | | $\Delta_{rel}$ (%/year) | $M_{O3}$ (DU) | Before 2012 | | After 2012 | | $\Delta_{rel}$ (%/year) |
| | | | $T_{O3}$ | $T_{rel}$ | $T_{O3}$ | $T_{rel}$ | | | $T_{O3}$ | $T_{rel}$ | $T_{O3}$ | $T_{rel}$ | |
| | | | (DU/year) | | | | | | (DU/year) | | | | |
| 9–15 km | Winter | 39.22 | 0.02 | 0.0 | −3.73 | −9.5 | −9.6 | 42.15 | −0.20 | −0.5 | −9.90 | −23.5 | −23.0 |
| | Spring | 39.26 | *6.66 | *17.0 | −2.37 | −6.0 | −23.0 | 48.48 | 3.87 | 8.0 | ※−12.71 | ※−26.2 | −34.2 |
| | Summer | 24.09 | 2.92 | 12.1 | 0.02 | 0.1 | −12.0 | 24.50 | −1.65 | −6.8 | −1.77 | −7.2 | −0.5 |
| | Autumn | 20.61 | 0.49 | 2.4 | −1.57 | −7.6 | −10.0 | 16.05 | −1.07 | −6.7 | −4.35 | −27.1 | −20.5 |
| 3–9 km | Winter | 18.38 | *2.43 | *13.2 | 1.08 | 5.9 | −7.3 | 19.46 | 2.64 | 13.6 | 0.67 | 3.4 | −10.1 |
| | Spring | 22.77 | *7.43 | *32.6 | *2.97 | *13.0 | −19.6 | 27.30 | *7.68 | *28.1 | −5.34 | −19.6 | −47.7 |
| | Summer | 25.57 | *4.16 | *16.3 | 0.55 | 2.1 | −14.2 | 26.69 | −0.27 | −1.0 | ※−6.80 | ※−25.5 | −24.5 |
| | Autumn | 20.04 | *3.35 | *16.7 | ※−2.15 | ※−10.7 | −27.4 | 12.34 | −0.01 | −0.1 | ※−4.47 | ※−36.2 | −36.2 |
| 0–3 km | Winter | 9.55 | ※0.93 | ※9.7 | ※1.83 | ※19.2 | 9.4 | 3.87 | | | | | |
| | Spring | 15.94 | *4.10 | *25.7 | *2.90 | *18.2 | −7.5 | 5.46 | | | | | |
| | Summer | 20.38 | *3.91 | *19.2 | 0.39 | 1.9 | −17.3 | 4.57 | | | | | |
| | Autumn | 13.18 | *3.34 | *25.3 | −0.33 | −2.5 | −27.9 | 3.14 | | | | | |