# Peer review of "Long-term Variations in Ozone Levels in the Troposphere and Lower Stratosphere over Beijing: Observations and Model Simulations"

_Atmospheric Chemistry and Physics, 2019_

## Referee Comment (RC1) · Anonymous Referee #3 · 18 Apr 2020

This paper summarizes an effort to diagnose recent changes in tropospheric ozone over the North China Plain from balloon-borne measurements (ozonesondes). This analysis is coupled with output from the Chemical Lagrangian Model of the Stratosphere (CLaMS) to help identify contributions from stratospheric ozone to observed changes over time. The analysis is mostly straight-forward, but it is not clear to me if this paper amounts to a substantial contribution. Similar recent studies have diagnosed the observed long-term changes using different datasets and have gone into somewhat greater detail to elucidate the processes responsible and the significance of diagnosed trends. I believe this could be a more meaningful contribution with greater communication of its novelty throughout and a bit deeper dive on the mechanisms responsible for

observed changes. Some specific concerns are outlined below.

General Comments:

1. The trend quantification in Section 5 is quite underwhelming. It is void of any statistical significance testing, which is necessary to diagnose meaningful changes. Moreover, trends are diagnosed for relatively short time periods, which provides little confidence in the result and leads to overfitting where there is substantial year-to-year variability. Confidence intervals (e.g., 95 and 99 percent) would be especially helpful here to demonstrate the degree to which diagnosed long-term changes (and reversals from the first to second half of the time period) are meaningful. Without statistical evaluation here, clear conclusions cannot be made for the diagnosed trends and the use of the term "significant" throughout the paper is inappropriate.

2. There are at least two claims based on comparison of observations and CLaMS output that are not justified based on the analyses conducted. First, the authors claim at lines 124-125 that CLaMS overestimates transport from the stratosphere to the troposphere. This is based on comparing ozone concentrations in the model to that observed and assuming a certain missing control by tropospheric chemistry. Without additional analysis (or citations to other more thorough evaluation), I do not find this claim to be justified based on the analysis presented in this paper. Second, at lines 261-266 it is argued that a reduction in stratospheric ozone found near the ozonesonde location in CLaMS is a result of ENSO, but there are certainly several alternative explanations for this change that are not acknowledged. Notably, Beijing is near the climatological mean latitude of the tropopause break (the sharp discontinuity in tropopause altitude from tropics to extratropics). Latitudinal migrations of the tropopause break could result in Beijing being more on the tropical side in later years, thus less exposed to downwelling stratospheric air. The latter can certainly be evaluated using the CLaMS output.

3. There is substantial repetition in the discussion of the time-evolving role of NOx.

Interactive
comment
Namely, a succinct analysis and discussion is given and then followed shorlty after by a less clear rehashing of essentially the same points while pointing to other work – lines 151-155. Perhaps the authors intend to make a slightly different point, but this is not clear.

Specific Comments:

Lines 34-35: delete "which transports stratospheric ozone into the troposphere and tropospheric ozone into the lower stratosphere" as it repeats the previous words in this sentence. Also, this sentence is incomplete. What about the exchange of ozone between the stratosphere and troposphere?

Line 66: "relative" should be "previous"

Line 74: The accuracy and precision of the ozonesonde data should be listed here.

Line 86: It is not clear where the "D" comes from in the ASAD acronym. A quick Google search shows that this should be defined here as "A Selfcontained Atmospheric chemistry coDe (ASAD)". Please revise.

Lines 140-148 and Figure 5: Details on OMI data used belong in the data and methods section.

Figure 6: It is not clear what exactly is being done to/with the data. Are the time series based on a three-month average of monthly means or something else?
* * *

---

## Author Comment (AC1) · 23 Apr 2020

Thanks for all the comments and suggestions. We have carefully revised the manuscript according to these suggestions. Our point-to-point responses are listed below:

General Comments: 1. The trend quantification in Section 5 is quite underwhelming. It is void of any statistical significance testing, which is necessary to diagnose meaningful changes. Moreover, trends are diagnosed for relatively short time periods, which provides little confidence in the result and leads to overfitting where there is substantial year-to-year variability. Confidence intervals (e.g., 95 and 99 percent) would be especially helpful here to demonstrate the degree to which diagnosed long-term changes (and reversals from the first to second half of the time period) are meaningful. Without statistical evaluation here, clear conclusions cannot be made for the diagnosed trends and the use of the term "significant" throughout the paper is inappropriate. Reply: We add statistical significance testing in Section 5. Most ozonesonde trends in lower troposphere and mid-troposphere before 2012 passed the 95% significance criterion. Since there are fewer samples after 2012, some trends only passed the 90% significance criterion.

2. There are at least two claims based on comparison of observations and CLaMS output that are not justified based on the analyses conducted. First, the authors claim at lines 124-125 that CLaMS overestimates transport from the stratosphere to the troposphere. This is based on comparing ozone concentrations in the model to that observed and assuming a certain missing control by tropospheric chemistry. Without additional analysis (or citations to other more thorough evaluation), I do not find this claim to be justified based on the analysis presented in this paper. Second, at lines 261-266 it is argued that a reduction in stratospheric ozone found near the ozonesonde location in CLaMS is a result of ENSO, but there are certainly several alternative explanations for this change that are not acknowledged. Notably, Beijing is near the climatological mean latitude of the tropopause break (the sharp discontinuity in tropopause altitude from tropics to extratropics). Latitudinal migrations of the tropopause break could result in Beijing being more on the tropical side in later years, thus less exposed to downwelling stratospheric air. The latter can certainly be evaluated using the CLaMS output. Reply: We claim that CLaMS overestimates transport from the stratosphere to the troposphere based on not only the comparison between ozonesonde and CLaMS, but also a study by Konopka et al. (2019). Although the current transport scheme in CLaMS shows a good ability to represent transport of tracers in the stably stratified stratosphere, there are deficiencies in the representation of the effects of convective uplift and mixing due to weak vertical stability in the troposphere. We give more explanation here to make it clearer to understand. For the second comment,

tropopause could also be a reason for the reduction in ozone in UTLS. We add a citation (Chen et al., 2019) in which the tropopause trend across China is investigated. The result shows an upwards trend of tropopause in most part of China including the North China Plain. The uplifted tropopause result in the reduction of ozone in UTLS. Chen, X., Guo, J., Yin, J., Zhang, Y., Miao, Y., Yun, Y., Liu, L., Li, J., Xu, H., Hu, K. and Zhai, P.: Tropopause trend across China from 1979 to 2016: A revisit with updated radiosonde measurements, International Journal of Climatology, 39(2), 1117– 1127. https://doi.org/10.1002/joc.5866, 2019.

3. There is substantial repetition in the discussion of the time-evolving role of NOx. Namely, a succinct analysis and discussion is given and then followed shorlty after by a less clear rehashing of essentially the same points while pointing to other work – lines 151-155. Perhaps the authors intend to make a slightly different point, but this is not clear. Reply: We modified this part to make it clearer. First, we cite some studies which revealed the reduction of NOx in many places of China in recent years. Then, we use OMI data to show the long-term variation of tropospheric NO2 over Beijing. After talk about NOx, we were thinking to cite studies to show the change of other precursors. But now we move them to the Discussion and conclusion Section. In Section 4, we revised the description as "The variations in the precursors of tropospheric ozone have dominant roles in the long-term variability of tropospheric ozone. In recent years, the Chinese government has started to invest time and resources in controlling air pollution. A review of 20 years of air pollution control in Beijing (UN Environment, 2019) reported reductions in NOx during the period 2013–2017. A clear decreasing trend in NOx emissions has been observed since 2012 (van der A et al., 2017). Zheng et al. (2018) also reported that emissions of NOx in China decreased by 21% during the time period 2013–2017. Wang et al. (2019) reported that NOx emissions in eastern China decreased by âĹij25% from 2012 to 2016. Tropospheric NO2, one of the precursors of tropospheric ozone, has gradually decreased over Beijing in recent years (Vu et al. 2019). We use the tropospheric column of NO2 from OMI to discuss the influence of precursors on the long-term variation of tropospheric ozone in Beijing. The
deseasonalized tropospheric columns of NO2 measured by OMI from 2004 to 2018 are shown in Figure 5. Tropospheric NO2 was increasing from 2004 to 2010, especially in 2009, leading to the increase of ozone in lower and upper troposphere. As Chinese government start to control air pollutions, tropospheric columns of NO2 were in a condition of relatively large fluctuation in the period of 2010-2013. Tropospheric NO2 over Beijing experienced two major fluctuations in this period, as shown by Gaussian-weighted means. Then tropospheric NO2 was gradually decrease since 2013, result in the hiatus of ozone increase in lower and upper troposphere." In the discussion and conclusions section, the description is revised as "The Chinese government has taken action to reduce air pollution since 2012 and the precursors of ozone have decreased gradually in recent years (Vu et al., 2019; Zheng et al., 2018). We show the reduction in tropospheric NO2 by using OMI measurements. Other studies have also shown that the other O3 precursors have decreased in recent years in China, including not only NOx but also SO2 and VOCs (Ma et al., 2016; van der A et al., 2017; Li et al., 2017; UN Environment, 2019; Wang et al., 2019). These reduction in ozone precursors are considered to be the main reason for the hiatus in the increase in ozone in the troposphere, especially in the lower troposphere."

Specific Comments: Lines 34-35: delete "which transports stratospheric ozone into the troposphere and tropospheric ozone into the lower stratosphere" as it repeats the previous words in this sentence. Also, this sentence is incomplete. What about the exchange of ozone between the stratosphere and troposphere? Reply: This sentence is changed as "The exchange of ozone between the stratosphere and the troposphere is also important to bring ozone into troposphere (Dufour et al., 2010, 2015; Neu et al., 2014)".

Line 66: "relative" should be "previous" Reply: "relative" is replaced by "previous".

Line 74: The accuracy and precision of the ozonesonde data should be listed here. Reply: The mean difference in the ozone partial pressure between the IAP and ECC ozonesondes was <0.5 mPa in the troposphere and <1 mPa in the lower strato-

sphere. The correlation coefficients for profiles by IAP ozonesondes and the ECC are greater than 0.99 (Xuan et al., 2004). The total ozone columns measured by the IAP ozonesonde and the Brewer spectrophotometer were in good agreement with a relative difference of 6%. For the total ozone column, the relative difference and correlation coefficient between IAP ozonesonde and Brewer instrument were 6% and 0.94.

Line 86: It is not clear where the "D" comes from in the ASAD acronym. A quick Google search shows that this should be defined here as "A Selfcontained Atmospheric chemistry coDe (ASAD)". Please revise. Reply: We revise it as "A Selfcontained Atmospheric chemistry coDe (ASAD)".

Lines 140-148 and Figure5: Details on OMI data used belong in the data and methods section. Reply: We give more details on OMI data in Section 2.

Figure 6: It is not clear what exactly is being done to/with the data. Are the time series based on a three-month average of monthly means or something else? Reply: No, the data is still monthly. That means we have 3 samples each year for each season. To make it clearer, we revised it as "Figure 6. Trends of monthly mean column ozone (DU) from the ozonesonde observations (black) and CLaMS simulations (red) in four seasons. There are 3 monthly values in each year for each season."

---

## Referee Comment (RC2) · Anonymous Referee #1 · 25 Apr 2020

This paper presents ozonesonde observations of ozone in the troposphere and stratosphere in Beijing during 2002-2018. The data are analyzed with a stratospheric ozone tracer model and discussed qualitatively in the context of recent emission changes in Beijing and satellite NO2 data. The ozonesonde data provide valuable information on the ozone variations during this period of drastic changes in anthropogenic pollutants in Beijing and are an important contribution to ozone and climate research. I have some concerns on analysis and interpretation of the results. The paper can be accepted after these concerns have been addressed.

Major comments:

[Figure]

(1) A sudden drop in lower tropospheric ozone (< 3 km) after 2011 is surprising. It is inconsistent with satellite NO2 data shown in Fig 5 (and NOx emission inventory) which indicate gradual decrease in NOx emission after 2011. Recently reported surface measurements at two rural sites near Beijing (Shangdianzi and Gucheng) also did not observe sudden ozone drop around 2011/12 (Xu et al., 2020). I suggest comparing satellite observed tropospheric ozone to verify the sudden change observed in the present study. If no problem is found on data quality, the stepwise change is most likely due to change in large scale dynamics after 2011. The stratospheric model used in this study shows little change in stratospheric contribution to lower tropospheric ozone, but it may be the case that transport within the troposphere played a role. I suggest author add more analysis in this direction. For example, back trajectories can be calculated to see if there was change in transport from other parts of troposphere after 2011.

(2) The trend analysis can be improved; it is not clear why the trend calculation in the main text is different from the linear regression shown in the figures. In addition, the level of statistical significance in trend analysis should be provided.

(3) The lower tropospheric ozone in the present study appeared to have a small positive trend after the 2011 drop (Fig 3). This trend is not supported by author's contention that NOx reduction has decreased ozone. It is instead similar to surface ozone increase observed in many urban areas from the Ministry of Ecology and Environment network since 2013, which has been attributed to the nonlinear chemistry of ozone precursors (NOx emission decrease and VOC emission increase) and aerosol decrease, as well as being affected by meteorological variation (see for example, Li et al., 2019; Liu and Wang, 2020).

Minor comments:

Page 2, line 40-42, "Increasing surface ozone . . ." . Please note that recent studies have shown levelling off/decrease in surface ozone levels in rural areas of eastern China and in outflow of eastern China air masses (Xu et al., 2020; Wang et al., 2019).

Page 2, line 56, Consider modifying the statement "it is not known. . ." as it contradicts the author's earlier review of Dufour (2018) on the lower tropospheric ozone trend in NCP (which includes Beijing).

Page 2, line 100, Define "average percentage method", and clarify why a different (linear regression method) is used in the figures.

References

Xu, X., Lin, W., Xu, W., Jin, J., Wang, Y., Zhang, G., Zhang, X., Ma, Z., Dong, Y., Ma, Q., Yu, D., Li, Z., Wang, D. and Zhao, H., 2020. Long-term changes of regional ozone in China: implications for human health and ecosystem impacts. Elem Sci Anth, 8(1), p.13. DOI: http://doi.org/10.1525/elementa.409

Wang, T, Dai, J, Lam, KS, NanPoon, C and Brasseur, GP. 2019. Twenty-five years of lower tropospheric ozone observations in tropical East Asia: The influence of emissions and weather patterns. Geophys Res Lett 46. DOI: 10.1029/2019GL084459

Li, K., Jacob, D. J., Liao, H., Shen, L., Zhang, Q., and Bates, K. H.: Anthropogenic drivers of 2013-2017 trends in summer surface ozone in China, P Natl Acad Sci USA, 116, 422-427, 2019.

Liu, Y., and Wang, T.: Worsening urban ozone pollution in China from 2013 to 2017 – Part 1: The complex and varying roles of meteorology, Atmos. Chem. Phys. Discuss., 2020, 1-28, 10.5194/acp-2019-1120, 2020.

Liu, Y., and Wang, T.: Worsening urban ozone pollution in China from 2013 to 2017 – Part 2: The effects of emission changes and implications for multi-pollutant control, Atmos. Chem. Phys. Discuss., 2020, 1-27, 10.5194/acp-2020-53, 2020.

---

## Editor Comment (EC1) · Xavier Querol (Editor) · 26 Apr 2020

Dear authors,

Thanks a lot for your reply.

The referee made an important comment that should be replied at the begining of your reply because in my opinion it is very relevant:

The analysis is mostly straight-forward, but it is not clear to me if this paper amounts to a substantial contribution. Similar recent studies have diagnosed the observed long-term changes using different datasets and have gone into somewhat greater detail to

elucidate the processes responsible and the significance of diagnosed trends. I believe this could be a more meaningful contribution with greater communication of its novelty throughout and a bit deeper dive on the mechanisms responsible for observed changes

Xavier Querol

---

## Referee Comment (RC3) · Anonymous Referee #4 · 6 May 2020

The paper uses ozone soundings in eastern China from 2002 to 2020 to analyze trends in different altitudes, 0-3km, 3-9km and 9-15km. The authors conclude on different trends at these altitudes and particularly increasing trends before 2011-2012 in the troposphere and decreasing trends afterwards. In the lower stratosphere observations show a slight increase before 2012 and constant values afterwards and some "super-position" of lower tropospheric and stratospheric trends in the free troposphere. To explain potential reason for these trends they compare the data with the Chemical Lagrangian Model of the Stratosphere (CLAMS). Notably CLAMS it is able to simulate the stratospheric ozone distribution, but has no tropospheric chemistry. To do the comparison of trends they calculated the relative changes for each period and season

and selected the strongest changes of these tendencies (before and after 2011-2012). They compare the ozone tendencies before and after 2011-2012 in both, model and observations. They conclude, that the stratospheric impact also significantly has contributed to the trends in the free troposphere (They termed 'superposition layer'). They also conclude, that trend changes in the emission strengths are the key driver for ozone changes in the lower troposphere corresponding to NO2 observational trends.

The main problem with the paper, which I see is, that it does not use any robust statistical metrics or error estimate. Trends are evaluated over time periods of very few years (2002-2011/2012 and afterwards) and the year to year variability is high. The tendencies, which are shown and discussed remain vague. E.g. Fig 3a) shows a trend of observed O3 of zero DU after 2012 (9-15km), in the conclusions (l.274/275) the negative trend in the stratospheric dominated regime is mentioned. Also the criteria to define time periods of trend changes are not motivated and seem to differ in different plots (Fig.6).

The observations are interesting in some parts, but the most interesting part, which is the change of ozone in the 9-15km layer, remains unexplained and is not analyzed. The authors discuss some links with ENSO without providing additional analyses and make no link to the tropopause location or jet, tropical widening. Potential tropospheric circulation aspects could in principle also play a role changing tropospheric long range transport of air masses with high stratospheric ozone from non-local downward transport.

As such I do not know what the key finding of the paper is. If so, is it the stratospheric change of trend? Is it its impact on the free troposphere? Given the methods and the coarse analysis I don't see the manuscript meeting the standards of an ACP publication in its current form, although some observations are interesting.

Major comments:

As I said the data record is interesting, but the analysis is more than coarse. The

authors should at least provide some statistical valid metrics for the significance of trends.

1) There are no significance or error estimates of the 'trends' (the authors state, that the the time series is too short for this, which is weird, since the focus of the paper is on trends)

2) The selection of time intervals to calculate trends seems arbitrary and different in different altitudes. Criteria are not clear and seem to differ (Fig. 6).

3) They should also explain more clearly the role of meteorology when interpreting the seasonally resolved trends in the free troposphere (note that the whole manuscript does not contain any mentioning of the monsoon, convection, tropopause, jets).

4) They use the CLAMS model, which has no tropospheric chemistry to compare ozone (as mentioned correctly by the authors). How do the authors exclude potential changes of tropospheric ozone sources, circulation changes and long-range transport, which could potentially also lead to different variability and trends? The fact that an incomplete model sometimes agrees with observations, does not automatically exclude other processes, which are not included in the model, to explain the observed ozone tendencies.

---

## Author Comment (AC2) · 6 May 2020

Thanks for all the comments and suggestions. We have carefully revised the manuscript according to these suggestions. Our point-to-point responses are listed below:

Major comments: (1) A sudden drop in lower tropospheric ozone (< 3 km) after 2011 is surprising. It is inconsistent with satellite $NO_2$ data shown in Fig 5 (and $NO_x$ emission inventory) which indicate gradual decrease in $NO_x$ emission after 2011. Recently reported surface measurements at two rural sites near Beijing (Shangdianzi and Gucheng) also did not observe sudden ozone drop around 2011/12 (Xu et al., 2020). I

[Figure]

suggest comparing satellite observed tropospheric ozone to verify the sudden change observed in the present study. If no problem is found on data quality, the stepwise change is most likely due to change in large scale dynamics after 2011. The stratospheric model used in this study shows little change in stratospheric contribution to lower tropospheric ozone, but it may be the case that transport within the troposphere played a role. I suggest author add more analysis in this direction. For example, back trajectories can be calculated to see if there was change in transport from other parts of troposphere after 2011. Reply: We compared ozonesonde to OMI observation (Fig. A) as suggested. The data quality of OMI in troposphere is not as good as in stratosphere, especially in lower troposphere where there are often missing values in dataset. Even so, the sudden drop in the period of 2011-2012 are still found in middle troposphere and especially in UTLS. So, we believe the data quality of ozonesonde and the sudden drop we found are reliable. The reason for this sudden drop may mainly due to the changes in UTLS rather than NOx emission. Because the sudden drop is also found in CLaMS simulation which has no tropospheric ozone chemistry.

(2) The trend analysis can be improved; it is not clear why the trend calculation in the main text is different from the linear regression shown in the figures. In addition, the level of statistical significance in trend analysis should be provided. Reply: We checked every value of trend in the main text, and they are the same as the linear regression in the figures. We add statistical significance testing in the O3 and NO2 trends, most of them passed the 95% significance criterion. For the seasonal O3 trends, most ozonesonde trends in lower troposphere and mid-troposphere before 2012 passed the 95% significance criterion. Since there are fewer samples after 2012, some trends only passed the 90% significance criterion.

(3) The lower tropospheric ozone in the present study appeared to have a small positive trend after the 2011 drop (Fig 3). This trend is not supported by author's contention that NOx reduction has decreased ozone. It is instead similar to surface ozone increase observed in many urban areas from the Ministry of Ecology and Environment network

since 2013, which has been attributed to the nonlinear chemistry of ozone precursors (NOx emission decrease and VOC emission increase) and aerosol decrease, as well as being affected by meteorological variation (see for example, Li et al., 2019; Liu and Wang, 2020). Reply: We noticed that O3 trend is still positive after the 2011 drop, but it is much slower than before due to the reduction of NOx. However, there are other precursors which might be responsible for the small positive trend after the 2011 drop. Thanks for showing us the two papers (Li et al., 2019; Liu and Wang, 2020). We added them when we mentioned the possible reasons of meteorological variation in the discussion and conclusions.

Minor comments: Page 2, line 40-42, "Increasing surface ozone . . .". Please note that recent studies have shown levelling off/decrease in surface ozone levels in rural areas of eastern China and in outflow of eastern China air masses (Xu et al., 2020; Wang et al., 2019). Reply: We added the recent studies which show the decrease in surface ozone levels in next paragraph.

Page 2, line 56, Consider modifying the statement "it is not known. . ." as it contradicts the author's earlier review of Dufour (2018) on the lower tropospheric ozone trend in NCP (which includes Beijing). Reply: We modified this sentence.

Page 2, line 100, Define "average percentage method", and clarify why a different (linear regression method) is used in the figures. Reply: we explained the method. The method is used to remove the seasonality in the time series. As a result, we got deseasonalized O3 (black dots in figures). Linear regression method is applied on the deseasonalized O3 to get the trend of O3. These methods are used on different steps for different purpose.
* * *
[Figure]

**Fig. 1.** Figure A. Deseasonalized monthly mean partial columns of ozone over Beijing (black solid lines and dots) measured by OMI and the corresponding Gaussian-weighted means using a half-width of 12 months

---

## Author Comment (AC4) · 12 May 2020

Thanks for all the comments and suggestions. We already added the significance and error estimates. The criteria to define time periods in Fig. 6 are consistent with the time intervals in Fig. 3 and Fig. 4 in which sudden decrease is defined as the period in which the most significant decrease in Gaussian-weighted deseasonalized ozone was observed. The periods of sudden decrease are different in different altitudes, so the time intervals in Fig. 6 are different in different altitudes. In this paper, we focused on the changes of ozone trend which mainly caused by the change of emission and the sudden decrease in 2011-2012 which is largely related to the transport from

stratosphere. The other meteorological reason such as ENSO and tropopause might also related to ozone variation as we discussed in the conclusion section. However, they are not the main points of this paper and less important than emission and stratospheric transport. There may be many other meteorological factors like jet and tropical widening, but obviously it is impossible for anyone to investigate all of them in one single paper. We would like to deeper dive on other mechanisms in the future. As for the key finding of this paper, we think the dataset itself, the trends it revealed and the sudden decrease are the most innovative parts. Based on the only long-term observed ozonesonde data in North China Plain, we revealed the very interesting changes in tropospheric and lower-stratospheric ozone. We use $NO_2$ form OMI to show the influence of precursor on the change of trend, and we use CLaMS model to show the influence of stratospheric transport on the sudden decrease of ozone in 2011-2012. All of these make this paper an interesting and relatively complete story which we don't agree to call it "coarse". Does a good paper must contain complicated methods or revealed all possible mechanisms? We have carefully revised the manuscript according to these suggestions. Our point-to-point responses are listed below:

Major comments: As I said the data record is interesting, but the analysis is more than coarse. The authors should at least provide some statistical valid metrics for the significance of trends. 1) There are no significance or error estimates of the 'trends' (the authors state, that the the time series is too short for this, which is weird, since the focus of the paper is on trends)

Reply: we already added the significance and error estimates in the previous version of our paper.

2) The selection of time intervals to calculate trends seems arbitrary and different in different altitudes. Criteria are not clear and seem to differ (Fig. 6).

Reply: the selection of time intervals to calculate trends is indeed different in different altitudes (Fig. 6.). But the criteria are not arbitrary, they are consistent with the time
intervals in Fig. 3 and Fig. 4 in which sudden decrease is defined as the period in which the most significant decrease in Gaussian-weighted deseasonalized ozone was observed. The periods of sudden decrease are different in different altitudes, so the time intervals in Fig. 6 are different in different altitudes. We gave a clearer description in Fig. 6.

3) They should also explain more clearly the role of meteorology when interpreting the seasonally resolved trends in the free troposphere (note that the whole manuscript does not contain any mentioning of the monsoon, convection, tropopause, jets).

Reply: after the analysis of the long-term trends and the sudden decrease of ozone. We gave the seasonal trends to show in which seasons the significant changes of ozone are observed. In this part, we think that the precursors are the most important factors for the ozone in the troposphere-dominated range, and the transport greatly affects the ozone in the stratosphere-dominated range. It doesn't mean that we can exclude the meteorological reasons such as monsoon, convection, tropopause and jets. They are not the key points of this paper and less important than precursors and transport. Actually, we mentioned ENSO and tropopause in the discussion and conclusions section. There may be many other meteorological factors which affect variation of ozone, but obviously it is impossible for anyone to investigate all of them in one single paper.

4) They use the CLAMS model, which has no tropospheric chemistry to compare ozone (as mentioned correctly by the authors). How do the authors exclude potential changes of tropospheric ozone sources, circulation changes and long-range transport, which could potentially also lead to different variability and trends? The fact that an incomplete model sometimes agrees with observations, does not automatically exclude other processes, which are not included in the model, to explain the observed ozone tendencies.

Reply: CLaMS is not used to simulate tropospheric ozone and to compare with

ozonesonde. We want to isolate and quantify the long-term trends caused by transport from the stratosphere and by tropospheric chemistry. There is no tropospheric chemistry in CLaMS which you think it is an incomplete model. However, it is the specialty makes it a very qualified model for this work (to isolate and quantify the trends caused by transport and by tropospheric chemistry). We did not exclude potential changes of tropospheric ozone sources, circulation changes, long-range transport and other unknown reasons, but they are not the key points of this paper. No paper can completely include all factors, especially some of them are still unknown. For this paper, we revealed the trends and the sudden decrease of ozone based on the rare ozonesonde dataset, and we found these changes in ozone are related to NO2 and transport. So far, it is a complete and interesting story. Other mechanisms can be investigated deeper and more complete in future works.

Please also note the supplement to this comment:
https://www.atmos-chem-phys-discuss.net/acp-2019-1145/acp-2019-1145-AC4-supplement.pdf

**Supplement:**

[Figure]

**Figure 3. Deseasonalized monthly mean partial columns of ozone over Beijing (black solid lines and dots) measured by the IAP ozonesonde and the corresponding Gaussian-weighted means using a half-width of 12 months (red curves). The blue solid lines estimate the linear trends (slope ∓ standard error) before and after the decrease in ozone during late 2011 and early 2012. The periods of decrease are represented by the time between the two blue dashed lines. The trends with "*" passed the 95% significance criterion.**

5

[Figure]

**Figure 4. Deseasonalized monthly mean partial columns of ozone over Beijing (black solid lines and dots) simulated by CLaMS and the corresponding Gaussian-weighted means using a half-width of 12 months (red curves). The blue solid lines estimate the linear trends (slope ∓ standard error) before and after the decrease in ozone during late 2011 and early 2012. The periods of decrease are represented by the time between the two blue dashed lines. The trends with "*" passed the 95% significance criterion.**

[Figure]

15

**Figure 5. Deseasonalized monthly mean columns of tropospheric NO₂ (black solid lines and dots) and the corresponding Gaussian-weighted means using a half width of 12 months (red curves). The blue lines estimate the linear trend before and after the sudden decrease of ozone during late 2011 and early 2012.**

20

**Table 1. Trends in the monthly mean partial column ozone (units: DU/year) at three altitudes in four seasons. The trends with "\*" passed the 95% significance criterion. The trends with "※" passed the 90% significance criterion.**

| | | Ozonesonde | | | | | | CLaMS | | | | | |
|---|---|---|---|---|---|---|---|---|---|---|---|---|---|
| | | $M_{O3}$ | Before 2012 | | After 2012 | | $\Delta_{rel}$ | $M_{O3}$ | Before 2012 | | After 2012 | | $\Delta_{rel}$ |
| | | | $T_{O3}$ | $T_{rel}$ | $T_{O3}$ | $T_{rel}$ | | | $T_{O3}$ | $T_{rel}$ | $T_{O3}$ | $T_{rel}$ | |
| 9–15 km | Winter | 39.22 | 0.02 | 0.0 | −3.73 | −9.5 | −9.6 | 42.15 | −0.20 | −0.5 | −9.90 | −23.5 | −23.0 |
| | Spring | 39.26 | *6.66 | *17.0 | −2.37 | −6.0 | −23.0 | 48.48 | 3.87 | 8.0 | ※−12.71 | ※−26.2 | −34.2 |
| | Summer | 24.09 | 2.92 | 12.1 | 0.02 | 0.1 | −12.0 | 24.50 | −1.65 | −6.8 | −1.77 | −7.2 | −0.5 |
| | Autumn | 20.61 | 0.49 | 2.4 | −1.57 | −7.6 | −10.0 | 16.05 | −1.07 | −6.7 | −4.35 | −27.1 | −20.5 |
| 3–9 km | Winter | 18.38 | *2.43 | *13.2 | 1.08 | 5.9 | −7.3 | 19.46 | 2.64 | 13.6 | 0.67 | 3.4 | −10.1 |
| | Spring | 22.77 | *7.43 | *32.6 | *2.97 | *13.0 | −19.6 | 27.30 | *7.68 | *28.1 | −5.34 | −19.6 | −47.7 |
| | Summer | 25.57 | *4.16 | *16.3 | 0.55 | 2.1 | −14.2 | 26.69 | −0.27 | −1.0 | ※−6.80 | ※−25.5 | −24.5 |
| | Autumn | 20.04 | *3.35 | *16.7 | ※−2.15 | ※−10.7 | −27.4 | 12.34 | −0.01 | −0.1 | ※−4.47 | ※−36.2 | −36.2 |
| 0–3 km | Winter | 9.55 | ※0.93 | ※9.7 | ※1.83 | ※19.2 | 9.4 | 3.87 | | | | | |
| | Spring | 15.94 | *4.10 | *25.7 | *2.90 | *18.2 | −7.5 | 5.46 | | | | | |
| | Summer | 20.38 | *3.91 | *19.2 | 0.39 | 1.9 | −17.3 | 4.57 | | | | | |
| | Autumn | 13.18 | *3.34 | *25.3 | −0.33 | −2.5 | −27.9 | 3.14 | | | | | |

---

## Author Response (AR1)

**Author's Response:**

Thanks for all the comments and suggestions. We have carefully revised the manuscript according to these suggestions.

Our point-to-point responses are listed below:

**Comments from Referee 1:**

Suggestions for technical corrections or reasons for rejection

This work reports observations of ozone in the troposphere and stratosphere obtained from launching ozonesondes in Beijing during 2002-2016. The data provide important information on the ozone variations during this period of drastic changes in the emissions of anthropogenic pollutants in the North China Plains (NCP). The data have been analysed with a stratospheric ozone tracer model and discussed qualitatively in the context of recent emissions information.

**I have two major concerns at this stage.**

(1) First, it is very surprising to see a huge drop in ozone below 3 km after 2011 which is not convincingly explained by emission changes, raising concern on potential problem with the observation data. I am saying the ozonesonde data has problem for certain, but I advise the author to give more information to eliminate readers' doubt on data quality – after all, this dataset is the basis of the paper. Did other ozone measurements in Beijing give similar result (i.e., sharp and step-wise ozone decrease after 2011)? Was there a sudden and persistent change in large-scale dynamics after 2011?

Author's response: We give more information about ozonesondes and relative references. The ozonesonde data has been proved reliable and used to validate satellite measurements (Bian et al., 2007) and model products (Wang et al., 2012).

We think the change in trend is mainly the result of decrease of ozone precursors. So, we add a longterm variation of tropospheric NO2 from OMI. The huge drop of ozone in 2011-2012 may attribute to the change of transport from stratosphere. Because CLaMS which has no tropospheric ozone chemistry also shows the huge drop.

There was no other ozone measurements in Beijing except satellite data which is not better than ozonesonde measurements below 3 km.

We discussed the possible dynamical factors (ENSO and tropopause) which may be related to the ozone change.

Author's changes in manuscript: Information about ozonesondes is at Lines 64-76. Added  $NO_2$  from OMI is at Lines 161-168. Sudden decrease caused by transport is at Lines 284-286. The discussion about ENSO and tropapause is at Lines 275-282.

(2) Second, more in-depth analyses of the data are needed. It is better to use a model with reasonable representations of tropospheric ozone chemistry and/or other chemical (e.g., satellite observations of ozone precursors) and meteorology data to better explain the ozone changes. While the second point can be addressed during the discussion stage, I encourage the author to do this now to boost the rigor of the analysis.

Author's response: We use the version 3 of Aura Ozone Monitoring Instrument (OMI) Nitrogen Dioxide (NO2) standard product to discuss the influence of precursors on the long-term variation of tropospheric ozone in Beijing. The result shows that the decrease of tropospheric NO2 plays an important role in the decrease of tropospheric ozone.

Author's changes in manuscript: Lines 161–168 and Figure 5.

**Other quick suggestions to improve readability:**

(1) One or two sentence about the site will be helpful.

Author's response: The ozonesondes were released from Beijing Observatory (39.81°N, 116.47°E; 31 m above sea level).

Author's changes in manuscript: Line 80.

(2) Give a brief introduction of the chemistry in CLaMS.

Author's response: We gave a brief introduction of the chemistry in CLaMS in Section 2.2. Author's changes in manuscript: Lines 84–88.

(3) Line 86, briefly describe how depersonalized ozone is achieved.

*Author's response:* We added a brief introduction about how depersonalized ozone is achieved. *Author's changes in manuscript:* Lines 113–116.

(4) Fig 1 seems to show seasonal (not deseasonalized) ozone?

06 Jul Jan-07 Jul Jan-1

an-03 Jul Jan-0

Author's response: No, it is deseasonalized ozone. It is clearer to see the trend after remove the seasonal variation of ozone (Figure A).

0.03 0.07 0.11 0.15 0.19 0.23 0.27 0.31 0.35 0.39 0.43 0.47 (ppmv)

Figure A. Monthly mean ozone mixing ratio (units: ppmv) over Beijing measured by the IAP ozonesonde.

-08 Jul Jan-10 Jul Jan-11 Jul Jan-12 Jul Jan-13 Jul Jan-14 Jul Jan-16 Jul Jan-16 Jul Jan-17 Jul Ja

(5) Figure 1b: Why there are no data simulated using the model during July 2008, January, and July 2013? Author's response: There was no ozonesonde data during July 2008 and January 2013. In July 2013, we got ozone mixing ratio but lost the information of balloon locations (latitudes and longitudes during the fly). CLaMS is unable to calculate ozone mixing ratio without the information of balloon locations. Author's changes in manuscript: Lines 78–79 and lines 508–509.

(6) Table 1: Data for 0-3 km simulated by CLaMS are missing.

Author's response: Since there is no tropospheric ozone chemistry in CLaMS, it is meaningless to give data by CLaMS in 0-3 km which is significantly smaller than data by ozonesondes.

**Comments from Referee 1 in the interactive discussion:**

This paper presents ozonesonde observations of ozone in the troposphere and stratosphere in Beijing during 2002-2018. The data are analyzed with a stratospheric ozone tracer model and discussed qualitatively in the context of recent emission changes in Beijing and satellite NO2 data. The ozonesonde data provide valuable information on the ozone variations during this period of drastic changes in anthropogenic pollutants in Beijing and are an important contribution to ozone and climate research. I have some concerns on analysis and interpretation of the results. The paper can be accepted after these concerns have been addressed.

**Major comments:**

(1) A sudden drop in lower tropospheric ozone (< 3 km) after 2011 is surprising. It is inconsistent with satellite NO2 data shown in Fig 5 (and NOx emission inventory) which indicate gradual decrease in NOx emission after 2011. Recently reported surface measurements at two rural sites near Beijing (Shangdianzi and Gucheng) also did not observe sudden ozone drop around 2011/12 (Xu et al., 2020). I suggest comparing satellite observed tropospheric ozone to verify the sudden change observed in the present study. If no problem is found on data quality, the stepwise change is most likely due to change in large scale dynamics after 2011. The stratospheric model used in this study shows little change in stratospheric contribution to lower tropospheric ozone, but it may be the case that transport within the troposphere played a role. I suggest author add more analysis in this direction. For example, back trajectories can be calculated to see if there was change in transport from other parts of troposphere after 2011.

Author's response: We compared ozonesonde to OMI observation (Fig. B) as suggested. The data quality of OMI in troposphere is not as good as in stratosphere, especially in lower troposphere where there are often missing values in dataset. Even so, the sudden drop in the period of 2011-2012 are still found in middle troposphere and especially in UTLS. So, we believe the data quality of ozonesonde and the sudden drop we found are reliable. The reason for this sudden drop may mainly due to the changes in UTLS rather than NOx emission. Because the sudden drop is also found in CLaMS simulation which has no tropospheric ozone chemistry.

Figure B. Deseasonalized monthly mean partial columns of ozone over Beijing (black solid lines and dots) measured by OMI and the corresponding Gaussian-weighted means using a half-width of 12 months

(2) The trend analysis can be improved; it is not clear why the trend calculation in the main text is different from the linear regression shown in the figures. In addition, the level of statistical significance in trend analysis should be provided.

Author's response: We checked every value of trend in the main text, and they are the same as the linear regression in the figures. We add statistical significance testing in the  $O_3$  and  $NO_2$  trends, most of them passed the 95% significance criterion. For the seasonal  $O_3$  trends, most ozonesonde trends in lower troposphere and mid-troposphere before 2012 passed the 95% significance criterion. Since there are fewer samples after 2012, some trends only passed the 90% significance criterion.

Author's changes in manuscript: We provided statistical significance in Figures 3-5 and in Table 1.

(3) The lower tropospheric ozone in the present study appeared to have a small positive trend after the 2011 drop (Fig 3). This trend is not supported by author's contention that NOx reduction has decreased ozone. It is instead similar to surface ozone increase observed in many urban areas from the Ministry of Ecology and Environment network since 2013, which has been attributed to the nonlinear chemistry of ozone precursors (NOx emission decrease and VOC emission increase) and aerosol decrease, as well as being affected by meteorological variation (see for example, Li et al., 2019; Liu and Wang, 2020).

**Author's response:** We noticed that  $O_3$  trend is still positive after the 2011 drop, but it is much slower than before due to the reduction of NOx. However, there are other precursors which might be responsible for the small positive trend after the 2011 drop. Thanks for showing us the two papers (Li et al., 2019; Liu and Wang, 2020). We added them when we mentioned the possible reasons of meteorological variation in the discussion and conclusions.

Author's changes in manuscript: Lines 307-308, 400-401, 271-273 and 469-471.

**Minor comments:**

(1) Page 2, line 40-42, "Increasing surface ozone ...". Please note that recent studies have shown levelling off/decrease in surface ozone levels in rural areas of eastern China and in outflow of eastern China air masses (Xu et al., 2020; Wang et al., 2019).

Author's response: We added the recent studies which show the decrease in surface ozone levels in next paragraph.

Author's changes in manuscript: Lines 46-48.

- (2) Page 2, line 56, Consider modifying the statement "it is not known..." as it contradicts the author's earlier review of Dufour (2018) on the lower tropospheric ozone trend in NCP (which includes Beijing). Author's response: We deleted this sentence. Author's changes in manuscript: Line 58.
- (3) Page 2, line 100, Define "average percentage method", and clarify why a different (linear regression method) is used in the figures.

Author's response: we explained the method. The method is used to remove the seasonality in the time series. As a result, we got deseasonalized O3 (black dots in figures). Linear regression method is applied on the deseasonalized O3 to get the trend of O3. These methods are used on different steps for different purpose.

Author's changes in manuscript: Lines 114-115.

**Comment from Referee 2:**

Suggestions for technical corrections or reasons for rejection

During the past years, many papers have been published focused on explaining both trends and contribution of tropospheric ozone in some areas. Although this is very interesting for understanding the ozone formation processes and to define strategies for reducing ozone pollution, providing tropospheric ozone estimations is still a challenge because tropospheric ozone is a secondary pollutant and the formation and destruction reactions of it are very complex and difficult to modelize. Besides, it's difficult to isolate the contribution of the different precursors sources. In this paper, Zhang Y. et al. show the trend of ozone observations in Beijing from 2010 to 2018 providing relevant information about the levels of ozone in this area, which has not been studied yet. However, from my point of view, this paper is not very innovative because the applied methodology is very similar to previous works.

I first provide some general suggestions/observations regarding the paper and then I list specific comments.

**GENERAL COMMENTS**

(1) This study is focused on the trends of observed O3 measured in Beijing. A very similar study was done by Wang et.al. (2012) and this manuscript refers to this paper several times. From my point of view this should be highlighted in the introduction and it should be clearer because the applied methodology is really similar. This manuscript could be an extension of the first one.

Author's response: We highlighted the extension of the work by Wang et. al. in the introduction and it is clearer now.

Author's changes in manuscript: Lines 56-61.

(2) The authors use different names or different nomenclature along the text to refer to the same things. Defining these concepts at the beginning of the paper should be more convenient to avoid repetition of the definition of the different concepts. For example the names of the different layers with heights are in lines 101, 104, 139, 150.

The definition of the month of the year of each season should appear at the beginning, e.g.line 105.

Author's response: We checked the nomenclatures, especially the layers and the seasons. We gave the definitions when they were firstly mentioned and deleted the repeated description.

**SPECIFIC COMMENTS**

**Introduction**

(1) Line 40: Cooper et.al references doesn't appear in the Reference list.

Author's response: Cooper, O. R., Parrish, D. D., Ziemke, J., Balashov, N. V., Cupeiro, M., Galbally, I. E., Gilge, S., Horowitz, L., Jensen, J. -F., Naik, V., Oltmans, S. J., Schwab, J., Shindell, D. T., Thompson, A. M., Thouret, V., Wang, Y., and Zbinden, R. M.: Global distribution and trends of tropospheric ozone: An observation-based review, Elem. Sci. Anthr., 2, 000029,

doi:http://doi.org/10.12952/journal.elementa.000029, 2014.
Author's changes in manuscript: Lines 338–341.

Data and model

(2) Line 65: The authors say that the data used was measured about once a week. Explain in detail when data was collected (add dates and hours) and which gaps with no data you have if it's possible.

Author's response: The ozone profiles have been observed about once a week since 2002 at 14:00 local time (06:00 UTC). But we don't have a fixed date in one week. In some intensive observation periods (e.g., 24 March to 10 April 2003), ozonesondes were launched every day. However, there was no observation (gaps in Figure 1) in two periods (July 2008 and January 2013). Author's changes in manuscript: Lines 77–79.

(3) Line 76: Why 40-year of CLaMS simulation?

Author's response: We have 40-year of CLaMS simulation, and we use only the period of 2002-2018 which is consistent with ozonesonde data.

(4) Line 82: According to this manuscript there is no photochemistry in CLaMS model, but in Wang et.al. (2012) the model is executed without ozone chemistry (CLaMS-PO3: passively transported ozone). The model could be executed with and without chemistry? This should be defined more clearly. The definition of the model should be included before the explanation of the configuration.

Author's response: There is comprehensive photochemistry in CLaMS in stratosphere. However, there are very simple reactions in troposphere. The model could be executed with and without ozone chemistry. To isolate and quantify the long-term trend caused by transport from the stratosphere, a CLaMS simulation without ozone chemistry in troposphere is considered in this paper.

Author's changes in manuscript: Lines 95–99.

(5) Line 96: Define here the concepts of lower troposphere (3-9 km), mid-troposphere (9-15 km) and UTLS (9-15 km). Doing that the authors can use these terms without including the height. This information is repeated several times in the current version of the manuscript.

Author's response: We removed the repeated description of 0-3, 3-9, 9-15km.

(6) Line 104-106: T"The CLaMS simulations in the mid-troposphere (3-9) km are much closer to the ozonesonde measurements (Fig.2b). CLaMs seems to overestimate the transport of ozone from the stratosphere to the troposphere, which is strongest during winter and spring".
Comment: If we analyze the figure 2b, spring and summer are the seasons with highest overestimation of O3 values.

Author's response: After recheck Figure 2b, it is sure that CLaMS overestimate ozone in spring and underestimate ozone in autumn. It is hardly to say overestimate or underestimate in winter and summer except few dots. So, it should be more precise to say "CLaMs overestimates the transport of ozone from the stratosphere to the troposphere, which is strongest in spring".

Author's changes in manuscript: Lines 138–139.

(7) Line 105: Avoid using "seems".

Author's response: It was changed to "CLaMS overestimates the transport of ozone from the stratosphere to the troposphere, which is strongest during winter and spring". Author's changes in manuscript: Lines 138–139.

- (8) Line 113-114: Delete heights.
   Author's response: Deleted.
   Author's changes in manuscript: Line 133.
- (9) Line 139: Delete (0-3 km altitude)
   Author's response: Deleted.
   Author's changes in manuscript: Line 181.
- (10) Line 142: Delete (9-15 km altitude)
   Author's response: Deleted.
   Author's changes in manuscript: Line 183.
- (11)Line 141: Are there some previous studies analysing the most frequent mesoscale patterns to corroborate this?

Author's response: I don't understand this question. I am not sure what in Line 141 was asking to be corroborated by previous studies.

Tropospheric ozone chemistry dominates the trends in the lower troposphere (0–3 km altitude) in summer and autumn. The contribution in CLaMS is so small here that any stratospheric influence can be neglected. We call this range the "troposphere-dominated range". By contrast, the stratospheric influence is dominant in the UTLS (9–15 km altitude) in winter and spring and the tropospheric contribution can be ignored. We call this range the "stratosphere-dominated range". All the other combinations of seasons and altitudes are a superposition of the troposphere- and stratosphere-dominated ranges and we call such combinations the "superposition range".

**(12) Line 150: Replace "9-15km" by "UTLS"**

Author's response: "9-15km" has been replaced by "UTLS". Author's changes in manuscript: Line 192.

(13)Line 156-157: The months of each season should be defined previously.

Replace "before and after the decrease" by "before and after 2012" to be consistent with Figure 3 and 4.
Authors use different nomenclature in the text that doesn't appear in Table 1. For example MO3, TO3, Trel *Author's response:* "Before and after the decrease" has been replaced by "before and after 2012". We used MO3, TO3, Trel in Table 1 which consistent with the text.
Author's changes in manuscript: Table 1.

(14) Line 171: Why > 20%? Give more detailed information.

**Author's response:**  $\Delta_{rel}$  varies from 0% to 47.7%, we want to show the most significant change of ozone. So we choose  $\Delta_{rel} > 20\%$ . One can also choose > 15%, 25% or 30%... depends on how significant the ozone change was.

(15) Line 182: Delete (3-9 km)

Author's response: Deleted. Author's changes in manuscript: Line 223.

- (16) Line 190: Replace "almost neutral" by "almost zero". Author's response: "Almost neutral" has been replaced by "almost zero". Author's changes in manuscript: Line 231.
- (17) Line 220: Replace "the increase in ozone has been controlled" by " has been moderated compared to..". What does "been controlled" mean?

Author's response: We replaced "the increase in ozone has been controlled" by "the increase in ozone has been moderated since 2012".

Author's changes in manuscript: Line 262.

(18) Line 229: Wrong references: Replace "Zheng" by "Zhang"

Author's response: The reference is "Zheng, B., Tong, D., Li, M., Liu, F., Hong, C. P., Geng, G. N., Li, H. Y., Li, X., Peng, L. Q., Qi, J., Yan, L., Zhang, Y. X., Zhao, H. Y., Zheng, Y. X., He, K. B. and Zhang, Q.: Trends in China's anthropogenic emissions since 2010 as the consequence of clean air actions, Atmos. Chem. Phys., 18(19), 14095–14111, doi: 10.5194/acp-18-14095-2018, 2018." Author's changes in manuscript: Line 497.

(19) Line 231: Wrong reference: Want et. al. 2019 doesn't appear in the reference section. Author's response: Wang, N., Lyu, X., Deng, X., Huang, X., Jiang, F., and Ding, A.: Aggravating O3 pollution due to NOx emission control in eastern China, Science of the total environment, 677, 732-744, doi:10.1016/j.scitotenv.2019.04.388, 2019.

Author's changes in manuscript: Lines 466-468.

(20) Line 234: Wrong reference: Diallo et.al (2018) is from 2019.
 Author's response: It was changed to "Diallo et.al (2019)".
 Author's changes in manuscript: Line 278.

**Figures**

(21) Figure 2: Adding time period and specifying the months for season as the same way as figure 5. Author's response: We gave the time period and months for seasons in Figure 2. Author's changes in manuscript: Lines 511–515. (22) Figure 3: It's not clear. Replace Mean (DU) by MO3, "Trend" by TO3 and 'Relative trend' by Trel. Add new columns in the table instead of using ',' to separate fields.

*Author's response:* We Replace Mean (DU) by MO3, "Trend" by TO3 and 'Relative trend' by Trel in Table 1 which consistent with the text.

Author's changes in manuscript: Table 1.

References

(23) Wang. et. al (2009b): Include this reference after Wang. et. al (2009a)

Author's response: Wang, T., Wei, X. L., Ding, A. J., Poon, C. N., Lam, K. S., Li, Y. S., Chan, L. Y. and Anson, M.: Increasing surface ozone concentrations in the background atmosphere of Southern China, 1994–2007, Atmos. Chem. Phys., 9(16), 6217–6227, doi:10.5194/acp-9-6217-2009, 2009a.

Wang, X. S., Li, J. L., Zhang, Y. H., Xie, S. D., and Tang, X. Y.: Ozone source attribution during a severe photochemical smog episode in Beijing, China. Sci. China Ser. B, 52, 1270–1280, doi:10.1007/s11426-009-0137-5, 2009b.

Author's changes in manuscript: Lines 474–476 and lines 482–484.

**Comments from Referee 3**

(1) Suggestions for technical corrections or reasons for rejection

General comment: the ozonesonde location is very near the latitude of the tropopause break, which means it experiences both high-altitude tropopause (i.e., tropical) and low-altitude tropopause (i.e., extratropical) environments throughout the year. One very likely contributing factor to the changes discussed throughout is the time-evolving location relative to the tropopause break and how that relates to i) the degree of stratosphere-to-troposphere transport influence, and ii) the diagnosed UTLS ozone trend. I would recommend the authors include a detailed tropopause analysis to accompany their ozone assessment in order to whittle down the sources of uncertainty in the interpretation of the cause(s) and significance of observed changes. This has implications for the analysis and discussion presented throughout the manuscript.

Author's response: Thanks for the suggestion. Tropopause analysis would be a good way to investigate the variation of stratosphere-to-troposphere transport. We discussed tropopause in the last section.

Author's changes in manuscript: Lines 275–282.

(2) lines 34-35: should also mention transport of tropospheric ozone into the lower stratosphere.

Author's response: The transport of tropospheric ozone into the lower stratosphere is mentioned in revised paper.

Author's changes in manuscript: Lines 33-35.

- (3) Line 50: "2008—2012", extra character/hyphen here Author's response: Yes. We deleted the extra character. Author's changes in manuscript: Line 51.
- (4) Lines 105-106: Also worth mentioning the potential influence of tropospheric chemistry at this level. Author's response: In the mid-troposphere, transport from stratosphere is the main resource of CLaMS ozone because of the lack of tropospheric ozone chemistry in the model. Author's changes in manuscript: Lines 135–137.
- (5) Lines 170-175: If statistical significance was not determined, then I recommend not using the term "significance". Instead, terms like "most apparent" or "largest" could be used to get the same points across without introducing confusion since the meaning of "significant" is more subjective here. The same applies for other places in the paper where "significant" is used.

Author's response: We replaced the "significant" by "apparent" and "largest". Author's changes in manuscript: Line 213.

**Comments from Referee 3 in the interactive discussion:** General Comments: (1) The trend quantification in Section 5 is quite underwhelming. It is void of any statistical significance testing, which is necessary to diagnose meaningful changes. Moreover, trends are diagnosed for relatively short time periods, which provides little confidence in the result and leads to overfitting where there is substantial year-to-year variability. Confidence intervals (e.g., 95 and 99 percent) would be espe cially helpful here to demonstrate the degree to which diagnosed long-term changes (and reversals from the first to second half of the time period) are meaningful. Without statistical evaluation here, clear conclusions cannot be made for the diagnosed trends and the use of the term "significant" throughout the paper is inappropriate.

Author's response: We add statistical significance testing in Section 5. Most ozonesonde trends in lower troposphere and mid-troposphere before 2012 passed the 95% significance criterion. Since there are fewer samples after 2012, some trends only passed the 90% significance criterion. Author's changes in manuscript: Table 1.

(2) There are at least two claims based on comparison of observations and CLaMS output that are not justified based on the analyses conducted. First, the authors claim at lines 124-125 that CLaMS overestimates transport from the stratosphere to the troposphere. This is based on comparing ozone concentrations in the model to that observed and assuming a certain missing control by tropospheric chemistry. Without additional analysis (or citations to other more thorough evaluation), I do not find this claim to be justified based on the analysis presented in this paper. Second, at lines 261-266 it is argued that a reduction in stratospheric ozone found near the ozonesonde location in CLaMS is a result of ENSO, but there are certainly several alternative explanations for this change that are not acknowledged. Notably, Beijing is near the climatological mean latitude of the tropopause break (the sharp discontinuity in tropopause altitude from tropics to extratropics). Latitudinal migrations of the tropopause break could result in Beijing being more on the tropical side in later years, thus less exposed to downwelling stratospheric air. The latter can certainly be evaluated using the CLaMS output.

Author's response: We claim that CLaMS overestimates transport from the stratosphere to the troposphere based on not only the comparison between ozonesonde and CLaMS, but also a study by Konopka et al. (2019). Although the current transport scheme in CLaMS shows a good ability to represent transport of tracers in the stably stratified stratosphere, there are deficiencies in the representation of the effects of convective uplift and mixing due to weak vertical stability in the troposphere. We give more explanation here to make it clearer to understand.

For the second comment, tropopause could also be a reason for the reduction in ozone in UTLS. We add a citation (Chen et al., 2019) in which the tropopause trend across China is investigated. The result shows an upwards trend of tropopause in most part of China including the North China Plain. The uplifted tropopause result in the reduction of ozone in UTLS. Chen, X., Guo, J., Yin, J., Zhang, Y., Miao, Y., Yun, Y., Liu, L., Li, J., Xu, H., Hu, K. and Zhai, P.: Tropopause trend across China from 1979 to 2016: A revisit with updated radiosonde measurements, International Journal of Climatology, 39(2), 1117–1127. https://doi.org/10.1002/joc.5866, 2019.

Author's changes in manuscript: Lines 138-140 and Lines 275-278.

(3) There is substantial repetition in the discussion of the time-evolving role of NOx. Namely, a succinct analysis and discussion is given and then followed shorly after by a less clear rehashing of essentially the same points while pointing to other work – lines 151-155. Perhaps the authors intend to make a slightly different point, but this is not clear.

Author's response: We modified this part to make it clearer. First, we cite some studies which revealed the reduction of NOx in many places of China in recent years. Then, we use OMI data to show the long-term variation of tropospheric NO2 over Beijing. After talk about NOx, we were thinking to cite studies to show the change of other precursors. But now we move them to the Discussion and conclusion Section. In Section 4, we revised the description as "The variations in the precursors of tropospheric ozone have dominant roles in the long-term variability of tropospheric ozone. In recent years, the Chinese government has started to invest time and resources in controlling air pollution. A review of 20 years of air pollution control in Beijing (UN Environment, 2019) reported reductions in NOx during the period 2013-2017. A clear decreasing trend in NOx emissions has been observed since 2012 (van der A et al., 2017). Zheng et al. (2018) also reported that emissions of NOx in China decreased by 21% during the time period 2013–2017. Wang et al. (2019) reported that NOx emissions in eastern China decreased by â Lij25% from 2012 to 2016. Tropospheric NO2, one of the precursors of tropospheric ozone, has gradually decreased over Beijing in recent years (Vu et al. 2019). We use the tropospheric column of NO2 from OMI to discuss the influence of precursors on the long-term variation of tropospheric ozone in Beijing. The deseasonalized tropospheric columns of NO2 measured by OMI from 2004 to 2018 are shown in Figure 5. Tropospheric NO2 was increasing from 2004 to 2010, especially in 2009, leading to the increase of ozone in lower and upper troposphere. As Chinese government start to control air pollutions, tropospheric columns of NO2 were in a condition of relatively large fluctuation in the period of 2010-2013. Tropospheric NO2 over Beijing experienced two major fluctuations in this period, as shown by Gaussianweighted means. Then tropospheric NO2 was gradually decrease since 2013, result in the hiatus of ozone increase in lower and upper troposphere." In the discussion and conclusions section, the description is revised as "The Chinese government has taken action to reduce air pollution since 2012 and the precursors of ozone have decreased gradually in recent years (Vu et al., 2019; Zheng et al., 2018). We show the reduction in tropospheric NO2 by using OMI measurements. Other studies have also shown that the other O3 precursors have decreased in recent years in China, including not only NOx but also SO2 and VOCs (Ma et al., 2016; van der A et al., 2017; Li et al., 2017; UN Environment, 2019; Wang et al., 2019). These reduction in ozone precursors are considered to be the main reason for the hiatus in the increase in ozone in the troposphere, especially in the lower troposphere."

Author's changes in manuscript: Lines 153-168, 269-274 and 304-308.

**Specific Comments:**

(1) Lines 34-35: delete "which transports stratospheric ozone into the troposphere and tropospheric ozone into the lower stratosphere" as it repeats the previous words in this sentence. Also, this sentence is incomplete. What about the exchange of ozone between the stratosphere and troposphere? Author's response: This sentence is changed as "The exchange of ozone between the stratosphere and the troposphere is also important to bring ozone into troposphere (Dufour et al., 2010, 2015; Neu et al., 2014)".

Author's changes in manuscript: Lines 33-35.

- (2) Line 66: "relative" should be "previous" *Author's response:* "relative" is replaced by "previous". *Author's changes in manuscript:* Line 67.
- (3) Line 74: The accuracy and precision of the ozonesonde data should be listed here.

Author's response: The mean difference in the ozone partial pressure between the IAP and ECC ozonesondes was sphere. The correlation coefficients for profiles by IAP ozonesondes and the ECC are greater than 0.99 (Xuan et al., 2004). The total ozone columns measured by the IAP ozonesonde and the Brewer spectrophotometer were in good agreement with a relative difference of 6%. For the total ozone column, the relative difference and correlation coefficient between IAP ozonesonde and Brewer instrument were 6% and 0.94.

Author's changes in manuscript: Lines 71-76.

- (4) Line 86: It is not clear where the "D" comes from in the ASAD acronym. A quick Google search shows that this should be defined here as"A Selfcontained Atmospheric chemistry coDe (ASAD)". Please revise. *Author's response: We revise it as "A Selfcontained Atmospheric chemistry coDe (ASAD)". Author's changes in manuscript: Line 87.*
- (5) Lines 140-148 and Figure5: Details on OMI data used belong in the data and methods section. Author's response: We give more details on OMI data in Section 2. Author's changes in manuscript: Lines 101-111.
- (6) Figure 6: It is not clear what exactly is being done to/with the data. Are the time series based on a threemonth average of monthly means or something else?

Author's response: No, the data is still monthly. That means we have 3 samples each year for each season. To make it clearer, we revised it as "Figure 6. Monthly mean column ozone (DU) from the ozonesonde observations (black) and CLaMS simulations (red) in four seasons. Trends of column are calculated before and after the sudden decrease of ozone in 2011-2012. There are 3 monthly values in each year for each season."

Author's changes in manuscript: Lines 536-538.

**Comments from Referee 4**

Suggestions for technical corrections or reasons for rejection The presented long term ozone sonde data record itself is of interest for the community. However, the analysis does not meet the scientific standards from my point of view.

The most obvious ones are below:

 There are no significance or error estimates of the 'trends' (the authors state, that the the time series is too short for this, which is weird, since the focus of the paper is on trends)

Author's response: we already added the significance and error estimates in the previous and present version of our paper.

Author's changes in manuscript: Figures 3-5 and Table 1.

(2) The selection of time intervals to calculated trends is rather arbitrary and different in different altitudes. No criteria are given and it is speculated on common reasons for this.

Author's response: The selection of time intervals to calculate trends is indeed different in different altitudes, but the criteria are not arbitrary. The whole time series is divided by the time of sudden decrease of ozone in Fig. 3 and Fig. 4. It is the time period of sudden decrease that varies with altitudes. The time period of sudden decrease is defined as the period in which the most significant decrease in Gaussianweighted deseasonalized ozone was observed. So, the time intervals are different in different altitudes dut to the different times of sudden decrease. The time periods in Fig. 6 are consistent with the time intervals in Fig. 3 and Fig. 4. We gave a clearer description in Fig. 6. We gave a clearer description in Fig. 6. Author's changes in manuscript: Lines 150–152. Figure 6.

(3) There is no link to the meteorology: Does e.g. the monsoon (or a change of the circulation) play a role? Author's response: Meteorology may play a role based on the premise that a sudden change of meteorology has been observed around 2012. We didn't find such change in monsoon or circulation so far. Even the related change of meteorology has been found, how can it influence ozone is still a complicated question which is impossible to comprehensively show in this paper. We add discussion about ENSO and tropopause in the discussion and conclusions section. There may be many other meteorological factors which affect variation of ozone, but they are not the key points of this paper and less important than precursors and transport. It is impossible for anyone to investigate all of them in one single paper. We will show the influence of meteorology in our future study when there is enough evidence to support the link between meteorology and ozone variation.

Author's changes in manuscript: Lines 275-282.

(4) They use CLaMS model, which has no tropospheric chemistry to compare ozone (as stated correctly by the authors). The model has no tropospheric chemistry, nor convection included. How can you use this for tropospheric ozone comparisons?

Author's response: CLaMS is not used to simulate tropospheric ozone and to compare with ozonesonde. We want to isolate and quantify the long-term trends caused by transport from the stratosphere and by tropospheric chemistry. No tropospheric chemistry in CLaMS makes it a very qualified model for this work.

(5) Since CLaMS has no tropospheric ozone chemistry they conclude, that differences between the CLaMS ozone variability and observed ozone is driven by stratospheric processes. They do not discuss any reason for stratospheric transport changes or meteorology.

Author's response: Since CLaMS has no tropospheric ozone chemistry, CLaMS simulation shows the result caused only by stratospheric transport. The difference between the CLaMS ozone variability and observed ozone shows the effect of tropospheric chemistry.

The reasons for the changes of stratosperhic transport or meteorology are complicated. We are also studying these reasons until a satisfactory result has been achieved. We discussed the possible reasons (ENSO and tropopause) in the last section.

Author's changes in manuscript: Lines 275-282.

(6) The do not discuss other ozone sources, which are not included in the model (e.g. lightning NOx, the role of convection, which is not included by CLaMS, impact of wild fires and long range transport e.g. from siberia).

Author's response: We add the result of NO2 from OMI to discuss the influence of ozone precursors. Actually, the convection is included in CLaMS.

Author's changes in manuscript: Lines 161–168.

(7) They use the model and conclude on stratospheric influence without showing a unique stratospheric component (low humidity, other stratospheric species, etc.) to support their conclusion.

Author's response: CLaMS can simulate not only ozone but also other component. However, we have only ozone measurement by ozonesondes. So, there is no observation of other component can be compared with CLaMS. As for water vapor, we think it not a very good tracer to study stratospheric influence, because it may also come from descending of the Hadley cell.

Tao, M., Pan, L. L., Konopka, P., Honomichl, S. B., Kinnison, D. E., & Apel, E. C. (2018). A Lagrangian model diagnosis of stratospheric contributions to tropical midtropospheric air. Journal of Geophysical Research: Atmospheres, 123, 9764–9785. https://doi.org/10.1029/2018JD028696.

(8) They do not show any other chemical quantity, which is related to tropospheric (chemical) ozone change, at least satellite data could be compared.

Author's response: We add NO2, one of the most important ozone precursors which related to ozone change, in revised paper.

Author's changes in manuscript: Lines 161–168and Figure 5.

**Comments from Referee 4 in the interactive discussion:**

The paper uses ozone soundings in eastern China from 2002 to 2020 to analyze trends in different altitudes, 0-3km, 3-9km and 9-15km. The authors conclude on different trends at these altitudes and particularly increasing

trends before 2011-2012 in the troposphere and decreasing trends afterwards. In the lower stratosphere observations show a slight increase before 2012 and constant values afterwards and some "superposition" of lower tropospheric and stratospheric trends in the free troposphere. To explain potential reason for these trends they compare the data with the Chemical Lagrangian Model of the Stratosphere (CLAMS). Notably CLAMS it is able to simulate the stratospheric ozone distribution, but has no tropospheric chemistry. To do the comparison of trends they calculated the relative changes for each period and season and selected the strongest changes of these tendencies (before and after 2011-2012). They compare the ozone tendencies before and after 2011-2012 in both, model and observations. They conclude, that the stratospheric impact also significantly has contributed to the trends in the free troposphere (They termed 'superposition layer'). They also conclude, that trend changes in the emission strengths are the key driver for ozone changes in the lower troposphere corresponding to NO2 observational trends.

The main problem with the paper, which I see is, that it does not use any robust statistical metrics or error estimate. Trends are evaluated over time periods of very few years (2002-2011/2012 and afterwards) and the year to year variability is high. The tendencies, which are shown and discussed remain vague. E.g. Fig 3a) shows a trend of observed O3 of zero DU after 2012 (9-15km), in the conclusions (1.274/275) the negative trend in the stratospheric dominated regime is mentioned. Also the criteria to define time periods of trend changes are not motivated and seem to differ in different plots (Fig.6).

The observations are interesting in some parts, but the most interesting part, which is the change of ozone in the 9-15km layer, remains unexplained and is not analyzed. The authors discuss some links with ENSO without providing additional analyses and make no link to the tropopause location or jet, tropical widening. Potential tropospheric circulation aspects could in principle also play a role changing tropospheric long range transport of air masses with high stratospheric ozone from non-local downward transport.

As such I do not know what the key finding of the paper is. If so, is it the stratospheric change of trend? Is it its impact on the free troposphere? Given the methods and the coarse analysis I don't see the manuscript meeting the standards of an ACP publication in its current form, although some observations are interesting.

Author's response: Thanks for all the comments and suggestions.

We already added the significance and error estimates. The criteria to define time periods in Fig. 6 are consistent with the time intervals in Fig. 3 and Fig. 4 in which sudden decrease is defined as the period in which the most significant decrease in Gaussian-weighted deseasonalized ozone was observed. The periods of sudden decrease are different in different altitudes, so the time intervals in Fig. 6 are different in different altitudes.

In this paper, we focused on the changes of ozone trend which mainly caused by the change of emission and the sudden decrease in 2011-2012 which is largely related to the transport from stratosphere. The other meteorological reason such as ENSO and tropopause might also related to ozone variation as we discussed in the conclusion section. However, they are not the main points of this paper and less important than emission and stratospheric transport. There may be many other meteorological factors like jet and tropical widening, but obviously it is impossible for anyone to investigate all of them in one single paper. We would like to deeper dive on other mechanisms in the future.

As for the key finding of this paper, we think the dataset itself, the trends it revealed and the sudden decrease are the most innovative parts. Based on the only long-term observed ozonesonde data in North China Plain, we revealed the very interesting changes in tropospheric and lower-stratospheric ozone. We use NO2 form OMI to show the influence of precursor on the change of trend, and we use CLaMS model to show the influence of stratospheric transport on the sudden decrease of ozone in 2011-2012. All of these make this paper an interesting and relatively complete story which we don't agree to call it "coarse". Does a good paper must contain complicated methods or revealed all possible mechanisms?

We have carefully revised the manuscript according to these suggestions. Our point-to-point responses are listed below:

**Major comments:**

As I said the data record is interesting, but the analysis is more than coarse. The authors should at least provide some statistical valid metrics for the significance of trends.

 There are no significance or error estimates of the 'trends' (the authors state, that the the time series is too short for this, which is weird, since the focus of the paper is on trends)

Author's response: we already added the significance and error estimates in the present version of our paper.

Author's changes in manuscript: Figures 3-5 and Table 1.

(2) The selection of time intervals to calculate trends seems arbitrary and different in different altitudes. Criteria are not clear and seem to differ (Fig. 6).

Author's response: the selection of time intervals to calculate trends is indeed different in different altitudes (Fig. 6.). But the criteria are not arbitrary, they are consistent with the time intervals in Fig. 3 and Fig. 4 in which sudden decrease is defined as the period in which the most significant decrease in Gaussian-weighted deseasonalized ozone was observed. The periods of sudden decrease are different in different altitudes, so the time intervals in Fig. 6 are different in different altitudes. We gave a clearer description in Fig. 6.

Author's changes in manuscript: Lines 150–152 and Figure 6.

(3) They should also explain more clearly the role of meteorology when interpreting the seasonally resolved trends in the free troposphere (note that the whole manuscript does not contain any mentioning of the monsoon, convection, tropopause, jets).

Author's response: after the analysis of the long-term trends and the sudden decrease of ozone. We gave the seasonal trends to show in which seasons the significant changes of ozone are observed. In this part, we think that the precursors are the most important factors for the ozone in the troposphere-dominated range, and the transport greatly affects the ozone in the stratosphere-dominated range. It doesn't mean that we can exclude the meteorological reasons such as monsoon, convection, tropopause and jets. They are not the key points of this paper and less important than precursors and transport. Actually, we mentioned ENSO and tropopause in the discussion and conclusions section. There may be many other meteorological factors which affect variation of ozone, but obviously it is impossible for anyone to investigate all of them in one single paper.

Author's changes in manuscript: Lines 275-282.

(4) They use the CLAMS model, which has no tropospheric chemistry to compare ozone (as mentioned correctly by the authors). How do the authors exclude potential changes of tropospheric ozone sources, circulation changes and long-range transport, which could potentially also lead to different variability and trends? The fact that an incomplete model sometimes agrees with observations, does not automatically exclude other processes, which are not included in the model, to explain the observed ozone tendencies.

Author's response: CLaMS is not used to simulate tropospheric ozone and to compare with ozonesonde. We want to isolate and quantify the long-term trends caused by transport from the stratosphere and by tropospheric chemistry. There is no tropospheric chemistry in CLaMS which you think it is an incomplete model. However, it is the specialty makes it a very qualified model for this work (to isolate and quantify the trends caused by transport and by tropospheric chemistry). We did not exclude potential changes of tropospheric ozone sources, circulation changes, long-range transport and other unknown reasons, but they are not the key points of this paper. No paper can completely include all factors, especially some of them are still unknown. For this paper, we revealed the trends and the sudden decrease of ozone based on the rare ozonesonde dataset, and we found these changes in ozone are related to NO2 and transport. So far, it is a complete and interesting story. Other mechanisms can be investigated deeper and more complete in future works.

**Comments from editor:**

Dear authors, Thanks a lot for your reply. The referee made an important comment that should be replied at the begining of your reply because in my opinion it is very relevant: The analysis is mostly straight-forward, but it is not clear to me if this paper amounts to a substantial contribution. Similar recent studies have diagnosed the observed longterm changes using different datasets and have gone into somewhat greater detail to elucidate the processes responsible and the significance of diagnosed trends. I believe this could be a more meaningful contribution with greater communication of its novelty throughout and a bit deeper dive on the mechanisms responsible for observed changes.

Author's response: We noticed there are similar recent studies which diagnosed the long-term changes of ozone using different datasets. Most of these datasets are satellite observations or surface measurements. The data quality of satellite observations in troposphere are not as good as in stratosphere. Surface measurements are precise, but only surface  $O_3$  are measured. Compared to satellite data, our ozonesonde observations are more precise with much higher vertical resolution. Compared to surface measurements, we have the profiles from surface to ~30km. So, ozonesonde is the best dataset to investigated the ozone variation not only near surface but also in the whole troposphere and lower stratosphere. However, there are only a few stations have ozonesonde data and some of stations only measured for a short time. Beijing ozonesonde data is the longest observation (since 2002) of the ozone profile over the North China Plain. This dataset is once used in Wang et. al. (2012) to show positive trends in the period of 2002-2010 which raises many concerns (more than 60 citations). Now, we extend the observed time series where we used CLaMS to show the trends after 2010. Fortunately, we found a sudden decrease in 2011 which related to stratospheric transport, and we found negative trends in recent years which mainly due to the reduction of precursors. I think the dataset itself, the trends it revealed and the sudden decrease are the most innovative parts of this paper. As for the mechanisms, we discussed the two main reasons which is responsible for the change of trend and the sudden decrease. We use  $NO_2$  form OMI to show the influence of precursor on the change of trend, and we use CLaMS model to show the influence of stratospheric transport on the sudden decrease of ozone in 2011. There are other precursors which need more data and other possible reasons such as changes in meteorology, but they are not the main point of this paper. It is a huge task to investigate all the mechanisms responsible for the  $O_3$  change. We would like to deeper dive on other mechanisms in the future.

**Reference:**

Wang, Y., Konopka, P., Liu, Y., Chen, H., Müller, R., Plöger, F., Riese, M., Cai, Z. and Lü, D.: Tropospheric ozone trend over Beijing from 2002–2010: ozonesonde measurements and modeling analysis, Atmos. Chem. Phys., 12(18), 8389–8399, doi:10.5194/acp-12-8389-2012, 2012.

**A List of Major Comments:**

|                                 | Comments                               | Referees | Response                                       | Changes             |
|---------------------------------|----------------------------------------|----------|------------------------------------------------|---------------------|
| 1                               | significance or error estimates of the | 1, 3, 4  | added the significance and error               | Figures 3-5         |
|                                 | 'trends'                               |          | estimates                                      | and Table 1         |
| 2                               | time intervals to calculate trends     | 4        | the criteria are not arbitrary, they are       | Lines 150-          |
|                                 | seems arbitrary and different in       |          | based on the time of sudden decrease           | 152.                |
|                                 | different altitudes                    |          | which is different in different altitudes.     |                     |
|                                 |                                        |          | It is defined as the period in which the       |                     |
|                                 |                                        |          | most significant decrease in Gaussian-         |                     |
|                                 |                                        |          | weighted deseasonalized ozone was              |                     |
|                                 |                                        |          | observed.                                      |                     |
| 3                               | link to the meteorology: e.g. the      | 1, 4     | We add the discussion about ENSO               | Lines 275-          |
|                                 | monsoon, tropopause, a change of       |          | and tropopause in the last section.            | 282.                |
|                                 | the circulation                        |          | There may be many other                        |                     |
|                                 |                                        |          | meteorological factors, but they are           |                     |
|                                 |                                        |          | not the key points of this paper and           |                     |
|                                 |                                        |          | less important than precursors and             |                     |
|                                 |                                        |          | transport. We will study them in the           |                     |
|                                 |                                        |          | future, because it is impossible to            |                     |
|                                 |                                        |          | investigate all of them in one single          |                     |
|                                 |                                        |          | paper.                                         |                     |
| 4                               | CLaMS model has no tropospheric        | 4        | CLaMS is not used to simulate                  |                     |
|                                 | chemistry. How can you use this for    |          | tropospheric ozone and to compare              |                     |
| tropospheric ozone comparisons? |                                        |          | with ozonesonde. We want to isolate            |                     |
|                                 |                                        |          | and quantify the long-term trends              |                     |
|                                 |                                        |          | caused by transport from the                   |                     |
|                                 |                                        |          | stratosphere and by tropospheric               |                     |
|                                 |                                        |          | chemistry. No tropospheric chemistry           |                     |
|                                 |                                        |          | in CLaMS makes it a very qualified             |                     |
|                                 |                                        |          | model for this work.                           |                     |
|                                 | I advise the author to give more       | 1        | The ozonesonde data has been proved            | Ozonesonde          |
|                                 | information to eliminate readers'      |          | reliable, we give more information             | data is at          |
|                                 | doubt on data quality. Did other       |          | about ozonesondes and relative                 | Lines 64–           |
|                                 | ozone measurements in Beijing give     |          | references.                                    | 76. NO 2 |
|                                 | similar result? Was there a sudden     |          | We think the change in trend is mainly         | from OMI            |
|                                 | and persistent change in large-scale   |          | the result of decrease of ozone                | is at Lines         |
|                                 | dynamics after 2011?                   |          | precursors. So, we add a long-term             | 161–168.            |
|                                 |                                        |          | variation of tropospheric NO 2 from | Sudden              |
|                                 |                                        |          | OMI. The huge drop of ozone in                 | decrease            |

| _                          |  |                                        |                     |                                                |                     |
|----------------------------|--|----------------------------------------|---------------------|------------------------------------------------|---------------------|
|                            |  |                                        |                     | middle troposphere in 2011-2012 may            | caused by           |
|                            |  |                                        |                     | attribute to the change of transport           | transport is        |
|                            |  |                                        |                     | from stratosphere. Because CLaMS               | at Lines            |
|                            |  |                                        |                     | which has no tropospheric ozone                | 284–286.            |
|                            |  |                                        |                     | chemistry also shows the huge drop.            | ENSO and            |
|                            |  |                                        |                     | There was no other ozone                       | tropapause          |
|                            |  |                                        |                     | measurements in Beijing except                 | is at Lines         |
|                            |  |                                        |                     | satellite data which is not better than        | 275–282.            |
|                            |  |                                        |                     | ozonesonde measurements below 3                |                     |
|                            |  |                                        |                     | km.                                            |                     |
|                            |  |                                        |                     | We discussed the ENSO and                      |                     |
|                            |  |                                        |                     | tropopause which may be related to             |                     |
|                            |  |                                        |                     | the ozone change.                              |                     |
| +                          |  | It is better to use a model with       | 1                   | We add the NO 2 from OMI to discuss | The NO 2 |
|                            |  | reasonable representations of          |                     | the influence of precursors on the long-       | from OMI            |
|                            |  | tropospheric ozone chemistry           |                     | term variation of tropospheric ozone in        | is at Lines         |
|                            |  | and/or other chemical (e.g., satellite |                     | Beijing. The result shows that the             | 161-168             |
|                            |  | observations of ozone precursors)      |                     | decrease of tropospheric NO2 plays an          | and in              |
|                            |  | and meteorology data to better         |                     | important role in the decrease of              | Figure 5.           |
| explain the ozone changes. |  |                                        | tropospheric ozone. |                                                |                     |
| -                          |  | The lower tropospheric ozone in the    | 1                   | We noticed that O 3 trend is still  | Lines 307-          |
|                            |  | present study appeared to have a       |                     | positive after the 2011 drop, but it is        | 308, 400-           |
|                            |  | small positive trend after the 2011    |                     | much slower than before due to the             | 401, 271-           |
|                            |  | drop. It is similar to surface ozone   |                     | reduction of NOx. However, there are           | 273 and             |
|                            |  | increase observed in many urban        |                     | other precursors which might be                | 469-471.            |
|                            |  | areas from the Ministry of Ecology     |                     | responsible for the small positive trend       |                     |
|                            |  | and Environment network since          |                     | after the 2011 drop. Thanks for                |                     |
|                            |  | 2013, which has been attributed to     |                     | showing us the two papers (Li et al.,          |                     |
|                            |  | the nonlinear chemistry of ozone       |                     | 2019; Liu and Wang, 2020). We added            |                     |
|                            |  | precursors (NOx emission decrease      |                     | them when we mentioned the possible            |                     |
|                            |  | and VOC emission increase) and         |                     | reasons of meteorological variation in         |                     |
|                            |  | aerosol decrease, as well as being     |                     | the discussion and conclusions.                |                     |
|                            |  | affected by meteorological variation   |                     |                                                |                     |
|                            |  | (see for example, Li et al., 2019;     |                     |                                                |                     |
|                            |  | Liu and Wang, 2020).                   |                     |                                                |                     |
| ŀ                          |  | It should be clearer highlighted in    | 2                   | We highlighted the extension of the            | Lines 56-           |
|                            |  | the introduction that this manuscript  |                     | work by Wang et. al. in the introduction       | 61.                 |
|                            |  | is an extension of the work by         |                     | and it is clearer now.                         |                     |
|                            |  | Wang et al. (2012).                    |                     |                                                |                     |
| L                          |  |                                        |                     |                                                | 1                   |

| The authors use different names or     | 2 | We checked the nomenclatures            |            |
|----------------------------------------|---|-----------------------------------------|------------|
| different nomenclature along the       | - | especially the layers and the seasons   |            |
| text to refer to the same things       |   | We gave the definitions when they       |            |
| Defining these concents at the         |   | were firstly mentioned and deleted the  |            |
| beginning of the paper should be       |   | repeated description                    |            |
| more convenient to avoid repetition    |   | repeated description.                   |            |
| afthe definition of the different      |   |                                         |            |
| of the definition of the different     |   |                                         |            |
| concepts.                              |   |                                         | X: 100     |
| The authors claim that CLaMS           | 3 | We claim that CLaMS overestimates       | Lines 138- |
| overestimates transport from the       |   | transport from the stratosphere to the  | 140        |
| stratosphere to the troposphere. This  |   | troposphere based on not only the       |            |
| is based on comparing ozone            |   | comparison between ozonesonde and       |            |
| concentrations in the model to that    |   | CLaMS, but also a study by Konopka      |            |
| observed and assuming a certain        |   | et al. (2019). Although the current     |            |
| missing control by tropospheric        |   | transport scheme in CLaMS shows a       |            |
| chemistry. Without additional          |   | good ability to represent transport of  |            |
| analysis (or citations to other more   |   | tracers in the stably stratified        |            |
| thorough evaluation), I do not find    |   | stratosphere, there are deficiencies in |            |
| this claim to be justified based on    |   | the representation of the effects of    |            |
| the analysis presented in this paper.  |   | convective uplift and mixing due to     |            |
|                                        |   | weak vertical stability in the          |            |
|                                        |   | troposphere. We give more               |            |
|                                        |   | explanation here to make it clearer to  |            |
|                                        |   | understand.                             |            |
| There is substantial repetition in the | 3 | We reorganized the discussion.          | Lines 153- |
| discussion of the time-evolving role   |   |                                         | 168, 269-  |
| of NOx.                                |   |                                         | 274 and    |
|                                        |   |                                         | 304-308.   |
|                                        |   |                                         | 1          |

4

I

[revised manuscript text omitted]
 NO2 one degree around Beijing. 批注 [Y1]: This part has been modified and reorganized.

批注 [Y2]: We add more information about CLaMS chemistry.

[revised manuscript text omitted]

批注 [Y5]: We reorganized this part.

批注 [Y6]: We added discussion about possible dynamical reasons: ENSO and tropopause.

**设置了格式:** 字体: 10 磅, 英语(美国)

[revised manuscript text omitted]

---

## Author Response (AR2)

**Suggestions from Referee #1:**

I appreciate the authors' effort in addressing my comments and improving the manuscript. After looking at the satellite tropospheric ozone column given in the response, I am more puzzled by the large step change (decrease) in ozonesonde observed ozone after 2011-12. I did rough calculations from Fig. 3, multiple year averaged ozone dropped by ~22% in 0-3 km and ~13% in 3-9 km, which is about 17% in 0-9km. However, satellite observed tropospheric ozone showed only graduate decrease (not the large step change) which is consistent with the graduate decline in satellite NOx (and in the NOx emission from the emission inventory). Moreover, stratosphere transported ozone showed a sudden drop in 2011/12, bounced up afterward and then gradually decreased (see Fig 4). Therefore, more convincing explanations are needed for the large step change in the tropospheric ozone column since 2011-12. Otherwise, I wary about the integraty of the ozonesonde measurements.

**Reply:**

We appreciate Referee #1 who pays attention to the integraty of the ozonesonde measurements. There were small mistakes in OMI ozone columns given in the last response, and they have been modified as Figure B. In order to prove the integraty of the ozonesonde measurements, we compared the monthly mean partial columns of ozone measured by ozonesondes with the ones measured by OMI (Figure A). Since it is not a paper talks about the quality of OMI retrieval, we did not put Figures A and B in the manuscript.

The columns by ozonesonde are almost consistent with the columns by OMI in 3-9 and 9-15km (Figures A(a) and A(b)). However, in 0-3km, the columns by OMI are much smaller than the columns by ozonesonde (Figure A(c)). This is because that OMI retrieval near the earth's surface is not as well as in upper levels, not because that the ozonesonde is imprecise. Ozonesonde measurement is much more reliable than OMI measurement in 0-3km. Thus, our ozonesonde data has been used to validate satellite measurements (Bian et al., 2007; Cai et al., 2009).

Figure B shows the deseasonalized monthly mean partial columns of ozone measured by OMI. When we ignore the OMI columns in 0-3km due to its inaccuracy, the sudden drop of ozone during 2011-2012 still exist in 3-9 and 9-15 km (Figures B(a) and B(b)) where two datasets match each other (Figures A(a) and A(b)). The reason for the sudden drop of ozone may also come from the stratosphere. Because the CLaMS model simulations we used in this work doesn't have tropospheric ozone chemistry, but there are still sudden drops of ozone in 3-9 and 9-15 km in CLaMS.

So, we have confidence in the integraty of the ozonesonde measurements. Although the reasons for the sudden drop of ozone during 2011-2012 are not fully understood, this phenomenon was real. We hope the further studies by others or us will explain it.

[Figure]

*Figure A. Monthly mean partial columns of ozone over Beijing measured by OMI (red lines) and the ozonesondes (black lines).*

[Figure]

*Figure B. Deseasonalized monthly mean partial columns of ozone over Beijing (black solid lines and dots) measured by OMI and the corresponding Gaussian-weighted means using a half-width of 12 months*

**Suggestions from Referee #3:**

The authors have done a good job adjusting the messaging after the initial submission. I have only two minor technical corrections to suggest:

• Line 255: I believe "by ozonesonde" should be "by CLaMS" here.

• Table 1: the 95% significance label defined in the caption (a 5-point star) does not match that used in the table elements (an asterisk). In addition, I recommend clarifying the caption and/or table contents to better summarize the units of the trend estimates. In particular, I believe the trends for the relative changes should be in %/year rather than DU/year as indicated.

**Reply:**

• The words in Line 255 should be "by ozonesonde" not "by CLaMS". We replaced "they" by "the trends" to make this sentence clearer. (Line 252-253 in revised paper)

• Thanks for the referee's attention to details. It is because that the typeface in the caption is different from the typeface in the table, and we changed it in revised paper. We clarifying the different units of the trend in the revised table.

[revised manuscript text omitted]